# The TRPV4 channel links calcium influx to DDX3X activity and viral infectivity

P. Doñate-Macián [1,2], J. Jungfleisch[3], G. Pérez-Vilaró[3], F. Rubio-Moscardo[1], A. Perálvarez-Marín [2], J. Diez[3] & M.A. Valverde [1]

Ion channels are well placed to transduce environmental cues into signals used by cells to generate a wide range of responses, but little is known about their role in the regulation of RNA metabolism. Here we show that the TRPV4 cation channel binds the DEAD-box RNA helicase DDX3X and regulates its function. TRPV4-mediated $Ca^{2+}$ influx releases DDX3X from the channel and drives DDX3X nuclear translocation, a process that involves calmodulin (CaM) and the CaM-dependent kinase II. Genetic depletion or pharmacological inhibition of TRPV4 diminishes DDX3X-dependent functions, including nuclear viral export and translation. Furthermore, TRPV4 mediates $Ca^{2+}$ influx and nuclear accumulation of DDX3X in cells exposed to the Zika virus or the purified viral envelope protein. Consequently, targeting of TRPV4 reduces infectivity of dengue, hepatitis C and Zika viruses. Together, our results highlight the role of TRPV4 in the regulation of DDX3X-dependent control of RNA metabolism and viral infectivity.

[1] Laboratory of Molecular Physiology, Department of Experimental and Health Sciences, Universitat Pompeu Fabra, 08003 Barcelona, Spain. [2] Departament de Bioquímica i de Biologia Molecular, Unitat de Biofísica, Universitat Autònoma de Barcelona, 08193 Bellaterra, Spain. [3] Molecular Virology Group, Department of Experimental and Health Sciences, Universitat Pompeu Fabra, 08003 Barcelona, Spain. Correspondence and requests for materials should be addressed to M.A.V. (email: miguel.valverde@upf.edu)

Cells integrate and coordinate RNA metabolism and protein production in order to respond to different external stimuli[1–4]. Every step in the RNA metabolism is under specific modulation to ensure that transcription, splicing, RNA nuclear export and translation occur efficiently. Of the different protein families that participate in RNA regulation, RNA helicases are multifunctional proteins that catalyze different steps in the RNA metabolism[5]. RNA metabolism is also a target of RNA viruses that are completely dependent on host cell proteins for propagation. RNA helicases have been described as essential host factors to promote viral replication as well as cellular sensors that trigger innate immune responses[5–7]. DDX3X is a widely expressed DEAD-box RNA-binding helicase that is typically hijacked by several RNA viruses[8–10] and may also play an oncogenic role[11]. DDX3X participates in all aspects of RNA metabolism and thus requires shuttling between the cytoplasm and the nucleus[12,13]. However, the molecular mechanisms regulating the cellular distribution and function of DDX3X are poorly understood.

The transient receptor potential (TRP) cationic channels transduce environmental cues into signals used by cells to generate a wide range of responses[14]. A few TRP channels also respond to viral infection, modifying their expression[15,16]. However, the way in which the increased channel expression influences cell response to viral infection is unknown. The calcium-permeable nonselective transient receptor potential vanilloid 4 (TRPV4) cation channel is widely expressed, shows certain spontaneous activity, and generates intracellular $Ca^{2+}$ signals in response to several stimuli, including hypotonic cell swelling, mechanical forces, moderate heat and UVB radiation[17–21]. Consequently, TRPV4 participates in diverse physiological functions and pathological conditions, particularly those related to epithelia, endothelium and osteoarticular tissues[17,18,22] as well as in innate immunity[23,24].

In the present study, we set out to examine the interaction between the DDX3X helicase and the TRPV4 cation channel and how that interaction is relevant to different DDX3X functions, particularly those related to viral RNA translation and multiplication.

## Results

**TRPV4 cation channel interacts with DDX3X RNA helicase.** To identify novel cellular functions and signaling pathways involving TRPV4, we used an open ended yeast two-hybrid screening specific for membrane proteins[25] (Supplementary Fig. 1a, b). With the forty-four positive TRPV4 interactors identified (Supplementary Fig. 1c), we generated a TRPV4 protein-protein interactome attending to protein co-localization, gene co-expression, genetic interactions and domain conservation (Supplementary Fig. 2a). Among the functional gene ontology terms highlighted by the network (Supplementary Fig. 2b and Supplementary Table 1), those corresponding to immune system response and viral infection showed the highest $P$ value for the association (Supplementary Fig. 2c). Within this category we focused on DDX3X, an ATP-dependent RNA helicase from the DEAD-box helicase family[8,26] involved in multiple stages of the RNA metabolism, from transcription to translation[27]. Functionally, DDX3X plays fundamental roles in immune responses and in cancer[12]. Moreover, diverse RNA viruses hijack DDX3X[8–10] and $Ca^{2+}$ signaling[28–30], to multiply efficiently.

To confirm the interaction between TRPV4 and DDX3X, HEK293 cells overexpressing TRPV4-V5(tag) and DDX3X-Myc (tag) were used for co-immunoprecipitation assays (Fig. 1a). Using an antibody against the V5 epitope to immunoprecipitate TRPV4, we detected DDX3X as a coprecipitant of TRPV4 (Fig. 1a right lane). No co-immunoprecipitation was observed in cells

transfected solely with DDX3X (Fig. 1a left lane). Similar results were obtained in a reversed co-immunoprecipitation assay using an antibody against Myc to immunoprecipitate DDX3X and an antibody against the V5 epitope to detect TRPV4 (Supplementary Fig. 3a). Activation of TRPV4 by the synthetic agonists GSK1016790A[31] reduced the intensity of the band corresponding to co-immunoprecipitated DDX3X (Fig. 1a middle lane and Fig. 1c), suggesting a putative channel-state dependency of the interaction. To evaluate the physiological importance of this interaction in an endogenous environment, we used human Huh7 hepatocarcinoma cells that naturally express both DDX3X and TRPV4. Identical results were obtained when the endogenous TRPV4 was immunoprecipitated using an antibody against the channel protein and an anti-DDX3X antibody was used for detection of the helicase (Fig. 1b). The reduction of the TRPV4–DDX3X interaction following exposure to GSK1016790A was detected in both cells overexpressing tagged proteins and in cells endogenously expressing the channel and helicase (Fig. 1c).

Next, we tested whether DDX3X co-localized with TRPV4 in HeLa cells transfected with TRPV4-V5(tag) and DDX3X-Myc (tag) using confocal microscopy. To remove excessive cytosolic signal, cells were first permeabilized with digitonin, washed extensively to remove the cytosolic proteins, fixed, and then stained with the antibodies[32]. This procedure revealed a pool of DDX3X colocalized with TRPV4 (Fig. 1d). A clear overlapping of the signal plot profiles of TRPV4 and DDX3X was also detected under control conditions at the plasma membrane that was reduced upon treatment with GSK1016790A (Fig. 1e). Analysis of pixel co-localization of DDX3X-EGFP and TRPV4-ECFP reported a Pearson's correlation coefficient of ~0.8 under control conditions that was significantly reduced upon addition of GSK1016790A (Fig. 1f). The close interaction between TRPV4 and DDX3X was further tested by two independent experimental approaches. First, the proximity of TRPV4-CFP- and DDX3X-YFP-tagged proteins was tested by the fluorescent resonance energy transfer (FRET) technique[33]. Figure 1g shows a significant increase in maximal FRET efficiency in HeLa cells co-transfected with TRPV4-CFP and DDX3X-YFP compared to the condition in which TRPV4 had been activated with GSK1016790A or the control condition using soluble YFP. Second, we used an assay to pull down DDX3X followed by mass-spectrometry peptide identification (Supplementary Data 1). HeLa cells overexpressing TRPV4 and DDX3X were treated with isotonic control media or GSK1016790A. The assay generated 71 putative DDX3X partners present only in isotonic media, 93 when treated with GSK1016790A and 287 that were common to both conditions (Supplementary Fig. 3b). TRPV4 was only identified as putative DDX3X binding partner in isotonic control conditions, supporting the idea that TRPV4-DDX3X interaction is sensitive to channel activity (Supplementary Fig. 3c).

**TRPV4 activation triggers nuclear accumulation of DDX3X.** We queried whether the loss of TRPV4-DDX3X interaction following channel activation was the result of DDX3X shuttling to the nucleus, a process required for the DDX3X-mediated nuclear export of unspliced viral mRNAs[8] and the regulation of gene transcription[11]. HeLa cells, which do not express endogenous TRPV4[34], were transiently transfected with TRPV4 and DDX3X or DDX3X alone and were monitored under a confocal microscope to follow the localization of DDX3X. As previously shown[8], DDX3X showed a cytoplasmic location under control isotonic conditions (Fig. 2a). Activation of TRPV4 with physiological hypotonic stimuli (30% hypotonicity), or the more potent pharmacological agonist GSK1016790A promoted an

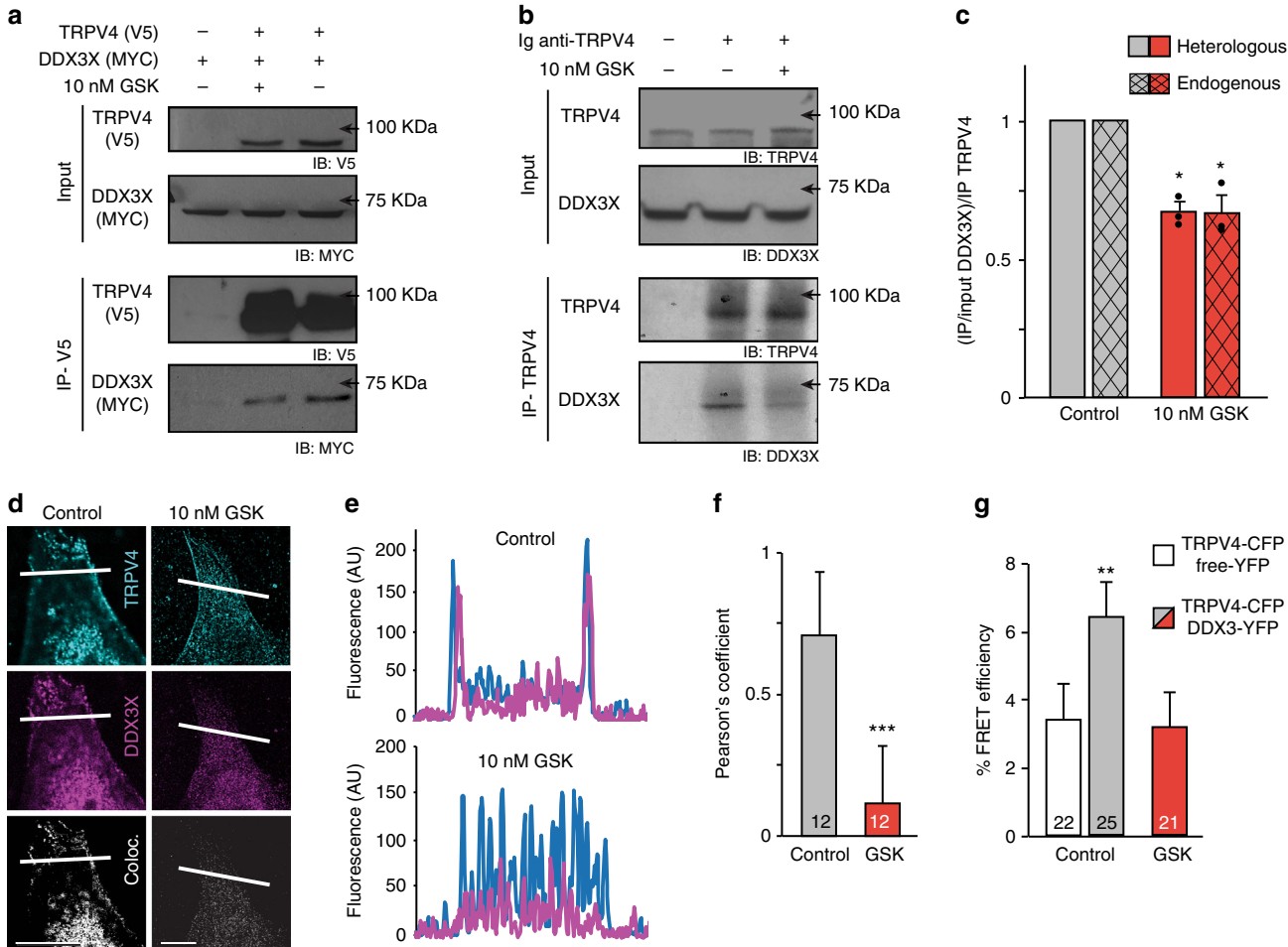

**Fig. 1** TRPV4 interacts with DDX3X. **a** Co-immunoprecipitation (Co-IP) of TRPV4-V5(tag) and DDX3X-Myc(tag) in HEK293 cells. **b** Co-IP of endogenously expressed TRPV4 and DDX3X in Huh7 cells. Uncropped westerns are shown in Supplementary Fig. 3d–f. **c** Quantification of immunoprecipitated DDX3X normalized by DDX3X input and immunoprecipitated TRPV4. Mean ± SEM of three Co-IPs of heterologously expressed or endogenously expressed TRPV4 and DDX3X proteins. *$P < 0.05$ ($P = 0.012$) two-tailed Student's $t$ test for heterologously and $P = 0.036$ for endogenously expressed proteins when compared control with GSK conditions. **d** Confocal immunofluorescence images of TRPV4 (cyan) and DDX3X (magenta) in HeLa cells overexpressing TRPV4-V5 and DDX3X-Myc. Co-localized pixels are shown in white in the merge panels. Scale bar = 10 μm. **e** Plot profile analysis performed using ImageJ software on each image along the white line shown on the panels. **f** Pearson's coefficient of the TRPV4 interaction with DDX3X under control conditions and upon channel activation with GSK1016790A. **g** FRET efficiency represented as the CFP increase during YFP photobleaching normalized to the initial CFP value measured in HeLa cells expressing TRPV4–CFP and DDX3X-YFP or free soluble YFP, and exposed to control condition or in the presence of 10 nM GSK1016790A. The number of cells analysed is indicated in each bar. Data are means ± SEM. ***$P < 0.001$ ($P = 0.0008$), two-tailed Student's $t$ test for **f** and **$P < 0.01$ one-way ANOVA followed by a Bonferroni post hoc test ($P = 0.007$) when comparing control TRPV4-CFP/free-YFP vs. any other condition (**g**). Bartlett's test for equal variances reported no significant differences ($P = 0.6$)

increase in intracellular $Ca^{2+}$ concentration (Supplementary Fig. 4a, b). Hypotonicity and GSK1016790A also triggered the accumulation of DDX3X in the nucleus (identified by the co-localization of DDX3X and DAPI nuclear staining) in cells expressing TRPV4 (Fig. 2a), but not in cells transfected with DDX3X alone (Supplementary Fig. 4c, d). Nuclear accumulation of DDX3X in response to GSK1016790A was prevented by the TRPV4 inhibitor HC067047[35] (Fig. 2a). Figure 2b shows the mean nuclear/total DDX3X intensity ratio under all conditions tested, which correlates with the magnitude of the TRPV4-mediated increases in intracellular $Ca^{2+}$ concentration (Supplementary Fig. 4a, b), i.e., the higher the $Ca^{2+}$ signal the higher the nuclear accumulation of DDX3X. To evaluate the physiological relevance of the TRPV4-induced nuclear shuttling of DDX3X in an endogenous environment, we used human bronchial epithelial (HBE)[36] and Huh7 hepatocarcinoma cells

that naturally expresses both DDX3X and TRPV4 (Fig. 1b). We observed that TRPV4 activation with GSK1016790A increased intracellular $Ca^{2+}$ concentration (Supplementary Fig. 4e, f) and promoted DDX3X shuttling to the nucleus (Fig. 2c, d). The participation of the endogenous TRPV4 in DDX3X nuclear shuttling was addressed following the silencing of TRPV4 with siRNA (Fig. 2c, d) or following pharmacological inhibition with HC067047 (Supplementary Fig. 4g). The efficiency of the TRPV4 knockdown in Huh7 cells was evaluated by quantitative PCR and calcium imaging. We estimated a TRPV4 knockdown of 75–90% that was maintained 96 h after transfection with siTRPV4, compared to control siRNA transfection (Supplementary Fig. 4h). Functional analysis of TRPV4 activity at different time points corroborated the efficiency of the TRPV4 silencing (Supplementary Fig. 4i, j). Both silencing of TRPV4 with siRNA (Fig. 2c, d) and TRPV4 inhibition with

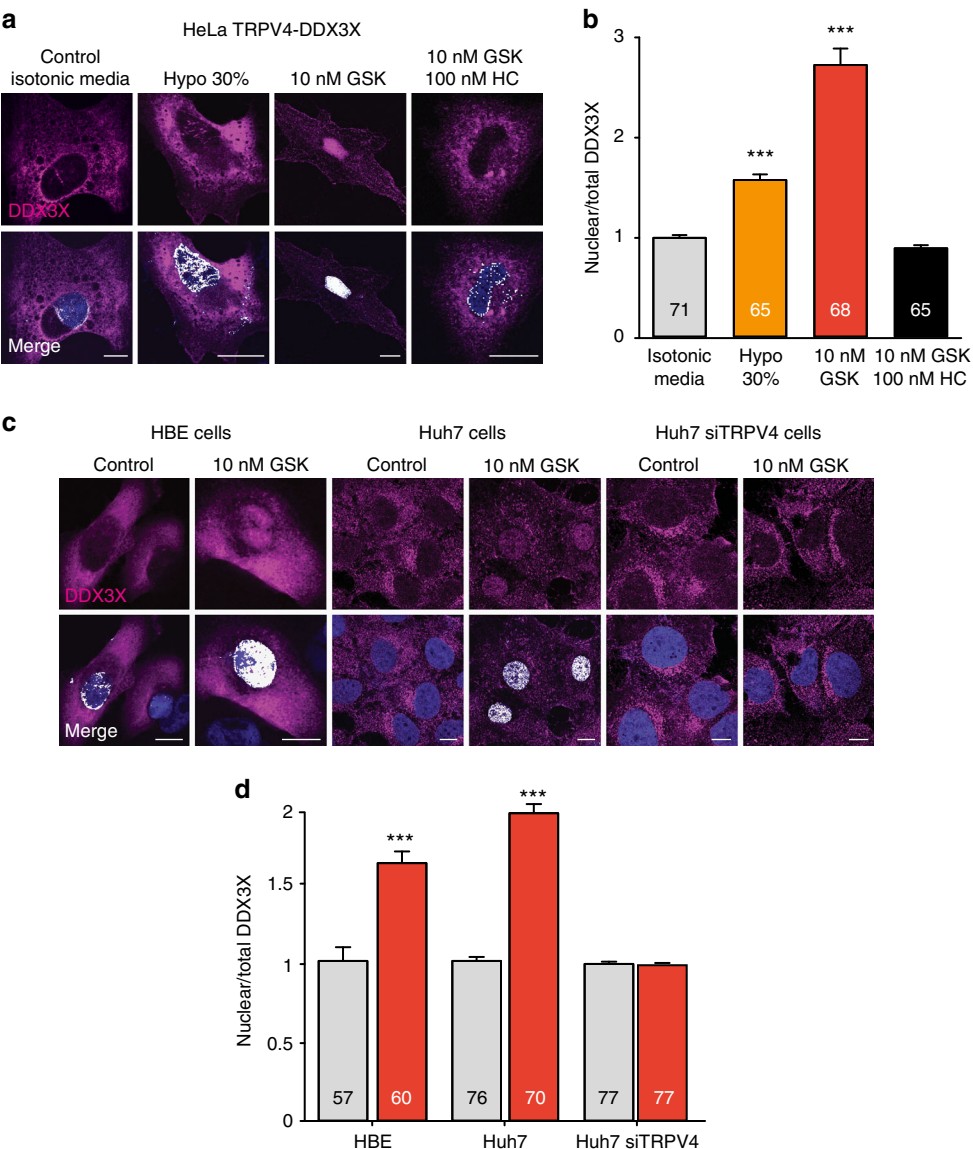

**Fig. 2** DDX3X shuttles to the nucleus upon TRPV4 activation. **a** Confocal immunofluorescence images of DDX3X (magenta) in HeLa cells overexpressing TRPV4-V5 and DDX3X-Myc. Nuclei were stained with DAPI (blue). Colocalization of DDX3X and nuclei is shown in white in the merge panels. **b** Mean nuclear/total DDX3X ratio determined under the experimental conditions indicated. **c** Confocal immunofluorescence co-localization (white) of DDX3X (magenta) with DAPI (blue) in HBE and Huh7 cells transfected with control or TRPV4 siRNAs. **d** Mean nuclear/total DDX3X ratio. The number of cells analysed is indicated in each bar. Data are means ± SEM. ***$P < 0.001$ when comparing isotonic control with any other condition as determined by non-parametric Krustal–Wallis followed by Dunn's post hoc test (**b**) and when comparing control vs. GSK conditions in HBE and Huh7 cells as determined by two-tailed Student's $t$ test (**d**). Scale bar = 10 μm

HC067047 (Supplementary Fig. 4g) prevented GSK1016790A-induced nuclear accumulation of DDX3X in Huh7 cells.

**DDX3X nuclear translocation requires Ca²⁺/CaM/CaM-kinaseII.** To confirm the Ca²⁺ dependency of DDX3X nuclear translocation, we removed extracellular Ca²⁺ before the addition of GSK1016790A (Supplementary Fig. 5a). Under these conditions, no nuclear translocation of DDX3X was observed (Fig. 3a, b). Moreover, exposure of HeLa cells to ionomycin (a Ca²⁺ ionophore), thapsigargin (an inducer of Ca²⁺ release from the endoplasmic reticulum) or Yoda1 (an agonist of the mechanosensitive Piezo1 channel[37]) did not induce DDX3X nuclear translocation (Fig. 3a, b) despite the marked increase in intracellular Ca²⁺ concentration triggered by all three stimuli

(Supplementary Fig. 5b–d). We also tested a dead-pore TRPV4 channel mutant (TRPV4-M680D)[38] that does not permeate Ca²⁺ when challenged with GSK1016790A (Fig. 3c). Under these circumstances, no translocation of DDX3X occurred, not even in the presence of ionomycin to force global Ca²⁺ influx (Fig. 3d, e). Together, these results suggested that a specific TRPV4-induced Ca²⁺ nano/microdomain generated in the proximity of the complex formed by TRPV4 and DDX3X is required to promote the translocation of DDX3X to the nucleus.

In order to identify the downstream pathway linking TRPV4 activation to DDX3X nuclear accumulation, HeLa cells overexpressing TRPV4 and DDX3X were exposed to inhibitors of calmodulin (W7 and Calmidazolium), calmodulin-dependent kinase II (KN93), tyrosine kinases (PP2), protein kinase G

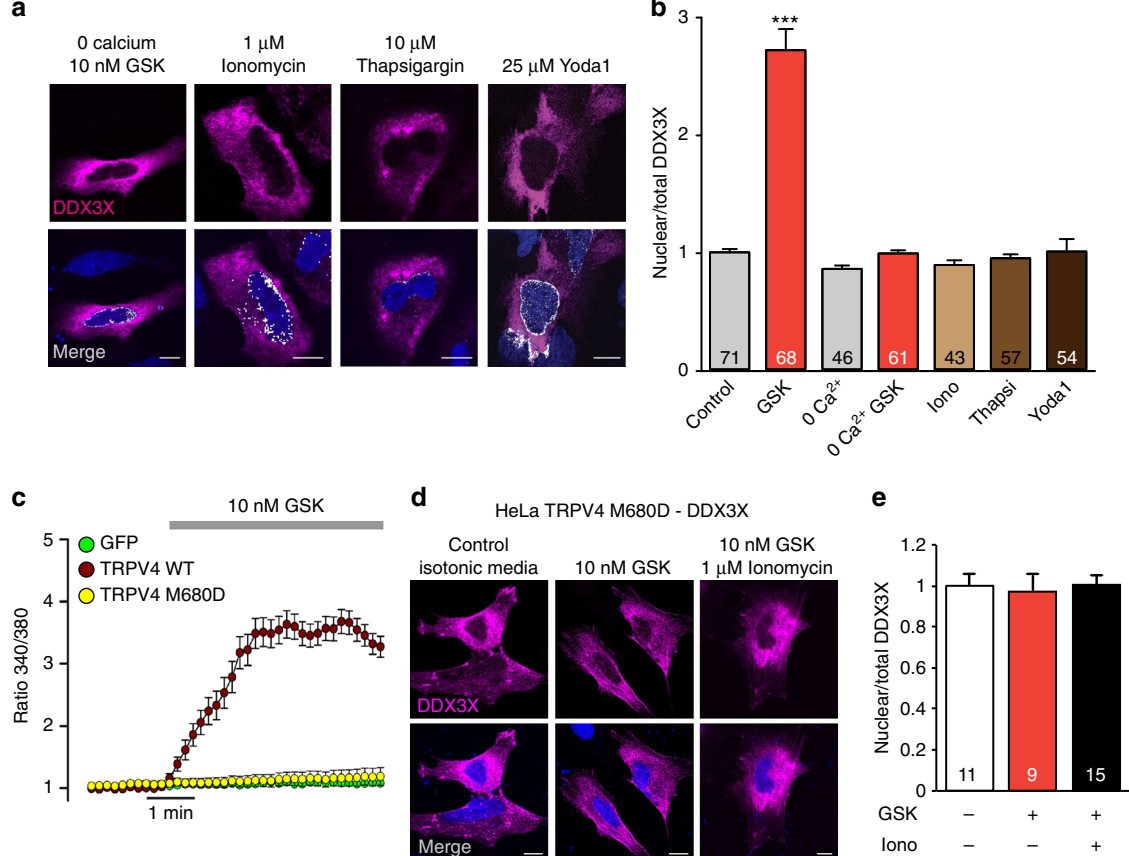

**Fig. 3** TRPV4-mediated Ca$^{2+}$ influx triggers DDX3X nuclear shuttling. **a** Confocal immunofluorescence images of DDX3X (magenta) in HeLa cells overexpressing TRPV4-V5 and DDX3X-Myc. Nuclei were stained with DAPI (blue). Colocalization of DDX3X and nuclei is shown in white in the merge panels. **b** Mean nuclear/total DDX3X ratio determined under the experimental conditions indicated. **c** Mean ± SEM of intracellular Ca$^{2+}$ signals (Fura-2 ratio) obtained in HeLa cells transfected with GFP ($n = 41$), TRPV4-WT ($n = 53$) or TRPV4-M680D ($n = 62$), and stimulated with 10 nM GSK1016790A. **d** Confocal immunofluorescence images of DDX3X (magenta) in HeLa cells overexpressing TRPV4-M680D and DDX3X-Myc. Nuclei were stained with DAPI (blue). Colocalization of DDX3X and nuclei is shown in white in the merge panels. **e** Mean nuclear/total DDX3X ratio determined under the experimental conditions indicated. The number of cells analyzed for each condition is indicated in each bar. The data are means ± SEM. ***$P < 0.001$ when comparing isotonic control with any other condition as determined by non-parametric Krustal–Wallis followed by Dunn's post hoc test (**b**, **e**). Scale bar = 10 μm

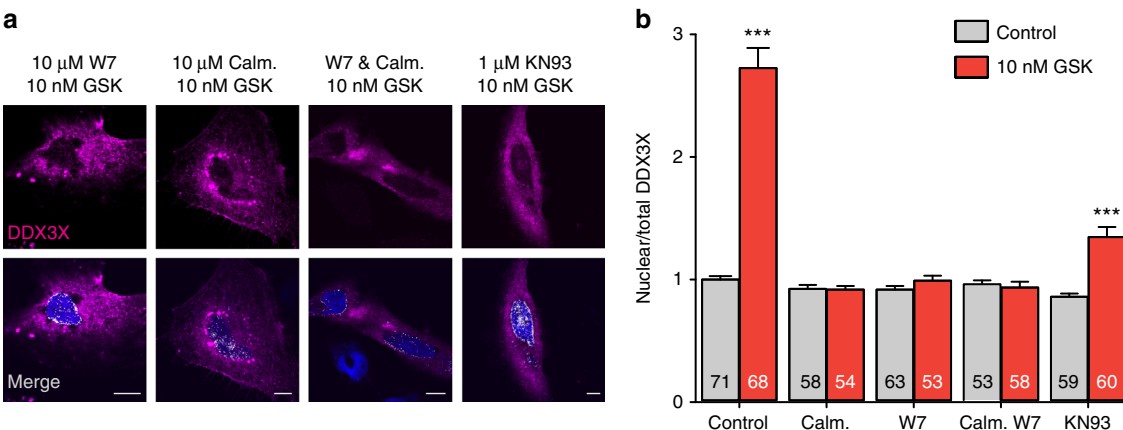

**Fig. 4** Calmodulin- and calmodulin kinase II-dependent nuclear accumulation of DDX3X. **a** Confocal immunofluorescence images of DDX3X (magenta) in HeLa cells overexpressing TRPV4-V5 and DDX3X-Myc. Nuclei were stained with DAPI (blue). Colocalization of DDX3X and nuclei is shown in white in the merge panels. **b** Mean± SEM nuclear/total DDX3X ratio determined under the experimental conditions indicated. The number of cells analyzed for each condition is indicated in each bar. The data are means ± SEM. ***$P < 0.001$ when comparing isotonic control with any other condition as determined by non-parametric Krustal–Wallis followed by Dunn's post hoc test. Scale bar = 10 μm

(KT5823), phospholipase A2 (AACoF3), protein kinase A (KT5720) or Calcineurin (Cyclosporin A). Excluding calmidazolium, all drugs allowed TRPV4 activation by GSK1016790A indistinguishable from the control conditions (Supplementary Fig. 6a). However, DDX3X translocation to the nucleus following TRPV4 activation was only abrogated in the presence of W7, calmidazolium and KN93 (Fig. 4a, b and Supplementary Fig. 6b–e). Thus, DDX3X nuclear translocation implicated the CaM/CaMKII pathway downstream of TRPV4 activation but unrelated to calmodulin or CaMKII shuttling to the nucleus as neither CaM (Supplementary Fig. 7a, b) nor CaMKII (Supplementary Fig. 7c, d) modified their cellular location following TRPV4 activation.

**TRPV4-DDX3X interaction regulates viral RNA translation.** DDX3X regulates gene expression at different levels, from transcription to translation[12]. To test whether the TRPV4-DDX3X interaction is relevant for DDX3X activity, we focused on well-known functions of DDX3X, particularly those related to viral RNA translation and multiplication. First, we used the yeast Brome mosaic virus (BMV) system as it has been shown that Ded1, the yeast orthologue of DDX3X, is required for translation of BMV RNA2[39]. Compared to the WT yeast strain, depletion of the TRPV4 orthologue in yeast, Yvc1 ($\Delta yvc1$), significantly inhibited translation of RNA2 and reduced levels of protein 2A (Fig. 5a, top) without significantly modifying the RNA2 levels (Fig. 5a, bottom) or the unrelated yeast PGK protein (Fig. 5a, middle). Mean reduction of protein 2A is shown in Fig. 5b. Transformation of the $\Delta yvc1$ yeast strain with human TRPV4 recovered RNA2 translation to levels similar to those obtained in WT cells (Fig. 5a, b), thereby confirming the implication of the channel in helicase-dependent viral RNA translation. Second, we examined the nuclear export and translation of unspliced HIV-1 genomic RNA (gRNA); processes that require DDX3X[8,40]. Viral gRNA is transcribed in the nucleus as pre-mRNA that either undergoes alternative splicing or, as in the case of the unspliced mRNA encoding the structural viral protein Gag, is directly exported from the nucleus. This nuclear export relies on the viral protein Rev and the host chromosome region maintenance 1 (CRM1) export receptor[7,41]. We used a method previously described that follows Rev-dependent nuclear export of a HIV-1 gRNA engineered to express the virus core structure protein GAG-CFP (to detect translation) and MS2-binding loops that will serve to detect viral RNA when co-expressed with MS2-YFP[41]. Despite the marked cytoplasmic location of Rev-mCherry in Huh7 cells, similar to HeLa cells overexpressing Rev-mCherry[41], Rev shuttles between the cytosol and nucleus supporting the nuclear export of HIV-1 gRNA (Fig. 5c). Inhibition of TRPV4 channel with HC067047 diminished gRNA nuclear export by ~22% (Fig. 5c, d), measured as the percentage of the difference in the total/nuclear MS2-YFP ratio between HC067047 and control conditions (i.e., inhibition of TRPV4 with HC067047 decreased cytosolic and increased nuclear MS2 signal, resulting in a reduced MS2 ratio compared to control condition, which is represented as negative $\Delta$ratio). Inhibition of TRPV4 also reduced translation by ~35%, expressed as a negative difference in the GAG-CFP intensity between HC067047 and control conditions (Fig. 5c–e). Further confirmation of the link between TRPV4 and DDX3X in the regulation of gRNA nuclear export and GAG translation was obtained by silencing TRPV4 and DDX3X with siRNA (Supplementary Fig. 8a). Silencing of either gene abrogated the effect of HC067047 (Fig. 5c–e), thereby suggesting that the reduction of gRNA export and translation induced by HC067047 depends on DDX3X. To test that inhibition of TRPV4 with HC067047 is related to Rev-mediated gRNA nuclear export we used a mutant

Rev-M10 that does not bind CRM1 and is retained in the nucleus[42]. As expected, expression of Rev-M10 was exclusively detected in the nucleus, whereas translation of GAG was totally abrogated (Supplementary Fig. 8b). Under these conditions, treatment with HC067047 did not modify MS2 location (Supplementary Fig. 8b, c) and confirmed that the effect of the channel inhibitor relied on the Rev-mediated nuclear export.

**TRPV4 responds to virus and participates in viral replication.** Several viruses such as HIV-1 induce cytosolic $Ca^{2+}$ signals originated from both depletion of intracellular stores and extracellular $Ca^{2+}$ influx[43,44]. Therefore, we evaluated the role of TRPV4 in the mechanism of viral infection. For that purpose, we use Huh7 cells amenable to infection by the emerging Zika virus (ZIKV). Huh7 cells were loaded with the calcium indicator Calbryte 520 AM and then exposed to control bathing solutions or to solutions containing ZIKV at a multiplicity of infection (MOI) of 1. ZIKV induced a slow increase in intracellular $Ca^{2+}$ concentration that was prevented by HC067047 (Fig. 6a). Exposure of Huh7 cells to the purified ZIKV envelope (E) protein (Supplementary Fig. 9), located at the surface of the viral particles, also generated an increase in intracellular $Ca^{2+}$ concentration (Fig. 6b) and nuclear accumulation of DDX3X that was inhibited by HC067047 (Fig. 6c, d). Viral infection triggers host innate immune responses characterized by increased expression of interferon β and other cytokines, a process in which DDX3X also participates[12,13]. We tested whether TRPV4 may also modulate this response by exposing Huh7 cells to GSK1016790A or HC067047 and quantifying the expression levels of tumor necrosis factor α, interleukin-2, interleukin 1β and interferon β. None of these changed under conditions that either activated or inhibited TRPV4 (Supplementary Fig. 10a). Infection with respiratory viruses has been shown to alter the expression of other thermosensitive TRP channels[15,16]. We also tested whether the expression of TRPV4 was modified by viruses that require DDX3X to multiply such as Dengue (DENV) and Hepatitis C (HCV) viruses. Neither virus modified TRPV4 mRNA levels in Huh7 cells (Supplementary Fig. 10b).

Lastly, we evaluated the role of TRPV4 in viral infectivity. Human Huh7 cells were exposed to ZIKV, DENV and HCV (all at MOI 0.01), and infectivity assessed by the activity of a luciferase reporter inserted in the viral RNA genomes. HC067047 reduced viral infectivity with half maximal effective concentration ($EC_{50}$) of 1.7 μM for DENV, 2 μM for HCV and 0.14 μM for ZIKV (Fig. 6e). The 50% cytotoxic concentration ($CC_{50}$) of HC067047 was 28 μM, resulting in a selectivity index ($CC_{50}/EC_{50}$) of 16, 13 and 187 for DENV, HCV and ZIKV, respectively. Another TRPV4 blocker structurally unrelated to HC067047, RN1734[45], also proved to be an effective inhibitor of HCV infection (Supplementary Fig. 11). Further confirmation for the role of TRPV4 in viral infection was obtained by exposing mouse embryonic fibroblast (MEF) generated from $Trpv4^{+/+}$ and $Trpv4^{-/-}$ mice to DENV. MEF $Trpv4^{-/-}$ cells showed a significant reduction in DENV infectivity with no changes in cellular viability (Fig. 6f).

**Discussion**
DDX3X is a DEAD-box RNA helicase ubiquitously expressed that participates in all aspects of RNA metabolism. DDX3X is implicated in tumorogenesis and is also the target of different RNA viruses that use the helicase as a host factor required for their replication[12,13]. RNA viruses account for a third of all emerging and re-emerging infections, such as those caused by Dengue, Zika, West Nile and Chikungunya viruses[46]. Therefore, understanding the regulation of RNA metabolism and their upstream

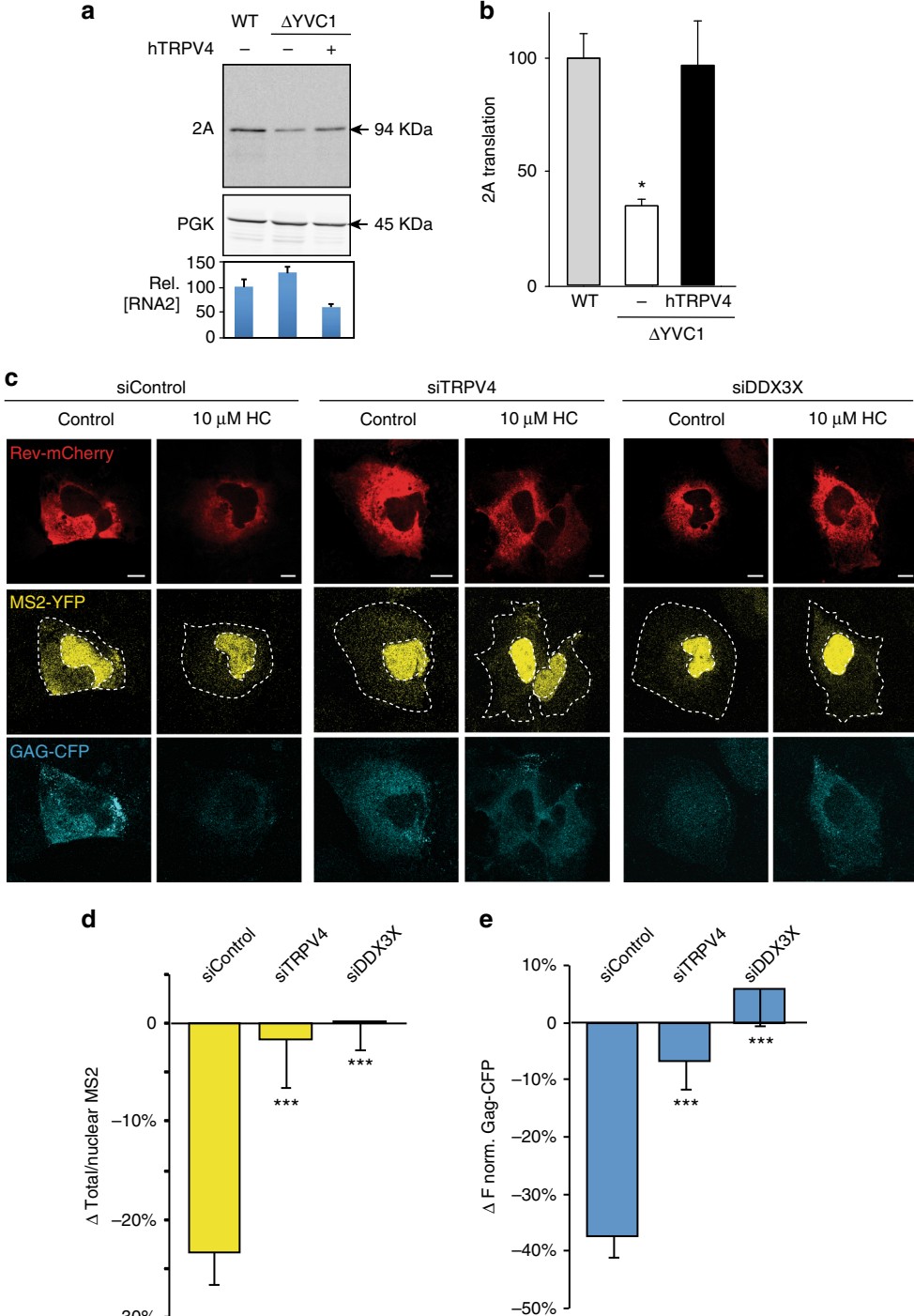

**Fig. 5** TRPV4 promotes nuclear export and translation of viral RNA. **a** Translation of viral RNA2 in yeast. Western blot analysis of protein 2A (top), protein PGK (middle) and qPCR quantification of viral RNA2 normalized to actin (lower) obtained in WT, Δ*Yvc1* and Δ*Yvc1* yeast transformed with hTRPV4. Differences in mean RNA2 values were not statistically significant ($P > 0.05$, $n = 3$) as determined by one-way ANOVA followed by a Bonferroni post hoc test when compared WT yeast with Δ*Yvc1* or Δ*Yvc1*+hTRPV4 ($P = 0.26$ and $P = 0.08$, respectively). **b** Quantification of RNA2 translation, calculated as the ratio of corrected protein 2a/RNA2. **c** Representative images of HeLa cells transfected with Rev-mCherry, RRE-gRNA and MS2-YFP, and stained with DAPI. **d** Quantification of nuclear export of gRNA, calculated as the difference in the total/nuclear MS2-YFP ratio between HC067047 and control conditions in cells transfected with siControl, siTRPV4 and siDDX3X. Mean (± SEM) of 67 siControl cells, 46 siTRPV4 cells and 47 siDDX3X analyzed from two independent experiments. **e** Quantification of Gag-CFP signal expressed as a negative difference in the GAG-CFP intensity between HC067047 and control conditions in cells transfected with siControl, siTRPV4 and siDDX3X. Cell contour marked by a dotted line. Mean ± SEM of 67 siControl cells, 46 siTRPV4 cells and 47 siDDX3X analyzed from two independent experiments. The data are means ± SEM. *$P < 0.05$ and ***$P < 0.001$ when comparing WT with any other condition (**b**) or siControl versus any other condition (**d**, **e**) as determined by one-way ANOVA followed by a Bonferroni post hoc test. Bartlett's test for equal variances reported significant differences in d and e ($P = 0.01$ and $P = 0.0001$, respectively). Scale bar = 10 μm

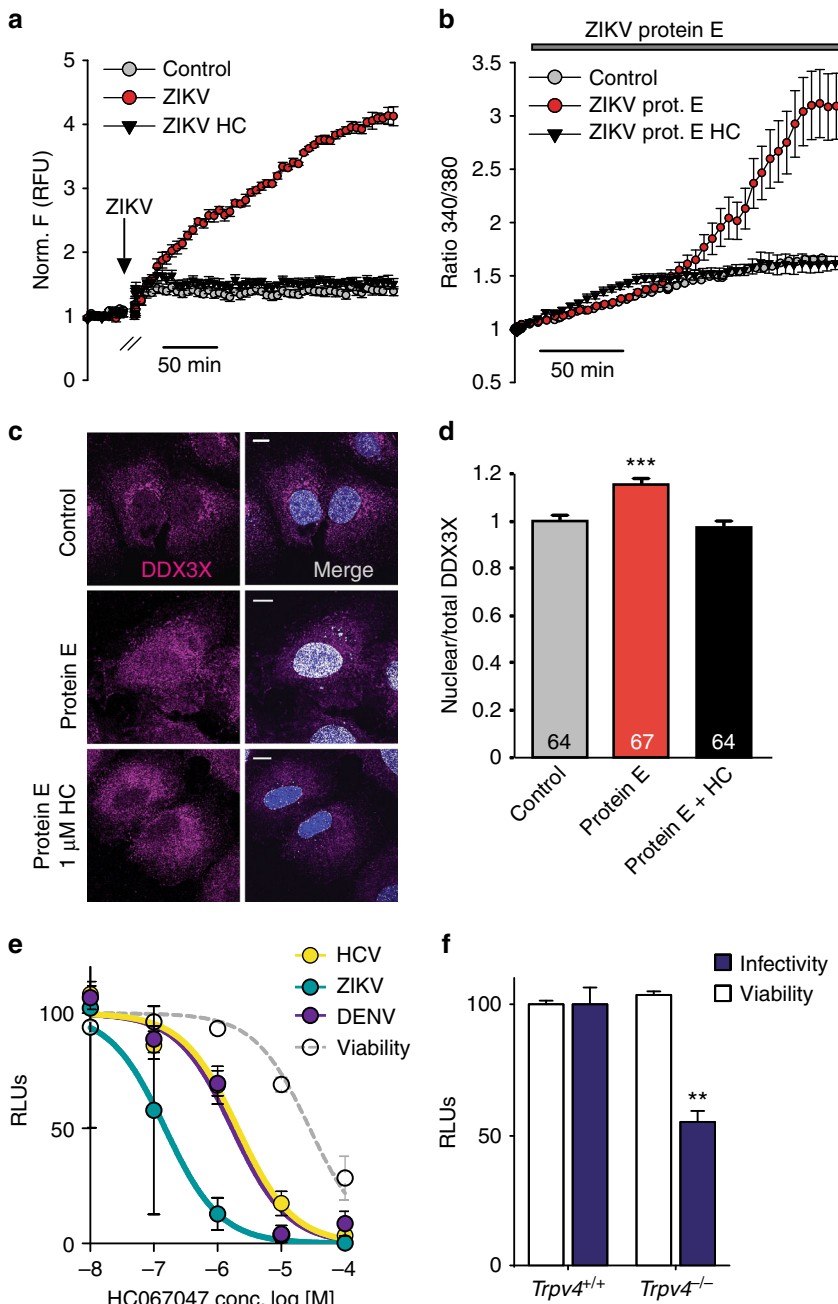

**Fig. 6** TRPV4 acts as a host factor for viral replication. **a** Mean (±SEM) of normalized intracellular $Ca^{2+}$ obtained from 3 experiments (with 70.000 cells analyzed in each experiment). Huh7 cells were loaded with the calcium indicator Calbryte 520 AM and exposed to control, ZIKV or ZIKV+10 μM HC067047. **b** Mean ( ± SEM) of normalized intracellular $Ca^{2+}$ signals (Fura-2 ratio) obtained in Huh7 cells exposed to control ($n = 38$), ZIKV (n = 48) or ZIKV+10 μM HC067047 ($n = 37$) conditions. **c** Confocal immunofluorescence images of DDX3X in Huh7 cells exposed to ZIKV in the presence or absence of HC067047. Nuclei were stained with DAPI (blue). Colocalization of DDX3X and nuclei is shown in white in the merge panels. **d** Mean nuclear/total DDX3X ratio determined under the experimental conditions indicated. **e** DENV, HCV and ZIKV infection in Huh7 cells treated with increasing concentrations of HC067047. The data are presented as Mean (±SEM) of three independent experiments at each HC067047 concentration. **f** DENV infection in $Trpv4^{+/+}$ and $Trpv4^{-/-}$ MEFs. Cellular viability was measured by CytoTox-Glo cytotoxicity assay and viral infectivity was measured as luciferase activity. The data are presented as Mean ( ± SEM) of three independent experiments. Data are means ± SEM. **$P < 0.01$ when comparing WT with any other condition as determined by one-way ANOVA followed by a Bonferroni post hoc test (**d**) or when comparing $Trpv4^{+/+}$ and $Trpv4^{-/-}$ mice as determined by two-tailed Student's $t$ test (**f**). $P = 0.001$ for infectivity and $P = 0.2$ for cell viability in **f**. Scale bar = 10 μm

signaling pathways may provide important insights into the way cells respond to environmental changes, including exposure to viruses.

During its life cycle, RNA is processed both in the nucleus and the cytoplasm. Accordingly, DDX3X, as other DEAD-box

helicases, normally shuttles between the nucleus and cytosol, with a preferential location to the latter compartment[8]. However, the mechanisms regulating such processes are not well understood. Through a combination of biochemical and imaging techniques, we have identified a close interaction between DDX3X and the

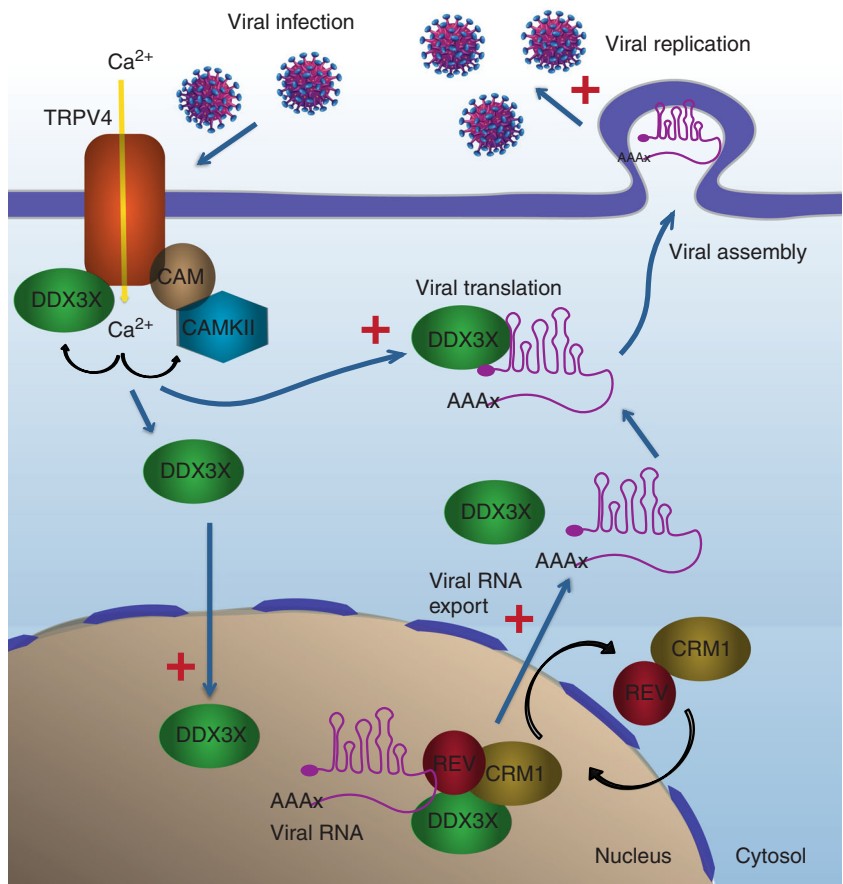

**Fig. 7** Model of the interaction between TRPV4 and DDX3X to regulate viral RNA metabolism. In the absence of external stimulation TRPV4 anchors both DDX3X and CaM. Following the activation of TRPV4 by viruses (or other physiological/pharmacological stimuli), DDX3X disengages from the channel and shuttles to the nucleus in a CaM/CaMKII-dependent process. In the nucleus, DDX3X facilitates the exit of unspliced viral RNA via de Rev/CRM1 export system. In the cytoplasm, DDX3X facilitates translation of viral RNA. Altogether, the interplay between TRPV4 and DDX3X promotes viral replication

plasma membrane TRPV4 cation channel. This interaction is lost upon channel activation, which results in increased intracellular $Ca^{2+}$ concentration and the nuclear accumulation of DDX3X (Fig. 7). The nuclear translocation of DDX3X in response to TRPV4 activation implies the participation of the $Ca^{2+}$/CaM/CaMKII pathway. The coupling of $Ca^{2+}$ signals to different cellular responses typically requires a specific timing and compartmentalization. The step $Ca^{2+}$ gradient around entry sites circumscribes the impact of $Ca^{2+}$ signals to distances up to a few hundred Å[47]. Thereby, downstream $Ca^{2+}$-binding proteins should be in close proximity to the source of $Ca^{2+}$. The TRPV4 cation channel may be well suited to act as a source of $Ca^{2+}$ and the scaffold that brings together all elements of this specific signaling module. CaM-mediated nuclear import is well conserved throughout evolution[48] and, both TRPV4[49,50] and DDX3X[51] bind CaM. The close proximity of all elements initiating this signaling cascade may be advantageous to rapidly activating its different elements. At present, we do not know whether a CaMKII-dependent phosphorylation of DDX3X is required for its nuclear transport or whether the effect of the kinase is at a different level in the amplification of the signaling cascade.

$Ca^{2+}$ signaling controls different cellular processes that are required by infecting viruses[28,44], thereby presenting an ideal target for viral proteins. Interestingly, the CaM signaling pathway is involved in Ebola, Dengue and Chikungunya infection[52–54]. The activation of TRPV4 by viral proteins links viral infection to the nuclear translocation of DDX3X, where it participates in the nuclear export of unspliced or partially spliced viral RNAs[8]. In addition, TRPV4 appears to be required for the helicase-dependent translation of viral proteins (Fig. 7). However, it is not clear yet how TRPV4 activity links to the viral replication of cytoplasmic-replicating viruses such as ZIKV, HCV and DENV that mainly depend on cytoplasmic DDX3X, nor the relevance of the TRPV4-DDX3X interaction in such process. Our study reveals that an ion channel, particularly well suited to sense the environment confers an additional regulatory layer in the tuning of RNA metabolism to changing environments. Altogether, our data propose TRPV4 as an interesting target to explore as therapeutic intervention in both DDX3X-dependent viral infections[55] and cancers in which DDX3X nuclear accumulation is associated to poor prognosis[56].

## Methods

**Membrane yeast two hybrid**. MYTH assay was previously described by Snider and cols[25]. NMY51 [*MATa his3delta200 trp1-901 leu2-3,112 ade2 LYS2::(lexAop)4-HIS3 ura3::(lexAop)8-lacZ (lexAop)8-ADE2 GAL4)*] (Dualsystems Biotech, P01401-P01429) was transformed with TRPV4-cUb-LexA-VP16 as bait. Western blot against LexA tag was used to confirm the correct expression of the generated baits. The MYTH assay was performed following the provider instructions (Dualsystems, P01401-P01429). Transformation efficiency of the human brain cDNA library into NMY51 carrying the bait constructs was higher than $2 \times 10^6$ colony formation units (CFU). Transformants were plated into 7,5 mM 3-amino-triazole (3AT) SD-LEU-TRP-HIS plates. After growth at 30 °C for 3–5 days, transformants were transferred into selective plates with same stringency but containing 5-bromo-4-chloro-3-indolyl-β,D-galactopyranoside (X-Gal). Prey plasmids DNA were isolated and transformed into DH5α *E. coli* competent cells to be sequenced. Isolated prey plasmids in frame with NubG and carrying a putative interaction partner were

transformed again into NMY51 strain containing the original bait and plated into selective plates with X-gal to confirm the interaction.

**Gene enrichment and network analysis**. Genes determined by the MYTH assay for TRPV4 (44 hits) were used to construct a network in Cytoscape with Genemania App. The network was constructed using the Genemania Human genome as a template and adding the 100 closest related genes based on co-localization, co-expression, physical interactions and domain conservation. Bingo Cytoscape app was used to find Go-Term categories statistically enriched within the TRPV4 network. P values of Go-terms were calculated with the hypergeometric test and corrected with the Benjamini & Hochberg FDR algorithm. Go-Terms were manually clustered by their function as cellular processes. Each cellular process includes a series of functionally related Go-Terms from Bingo analysis. TRPV4 partners associated to each cellular process were quantified and presented as a pie chart.

**Plasmids and molecular biology techniques**. pBT3-STE vector (P03233), control bait protein pTSU-APP, control prey pPR3-N (P01401-P01429) and cDNA library from human brain (P12227) were purchased from Dualsystems. Human TRPV4 full length was cloned into pPR3-N and pBT3-STE within SfiI sites. pcDNA3-TRPV4-WT, pcDNA3-TRPV4-CFP, TRPV4-V5 (tagged in the first extracellular loop) and calmodulin-CFP were previously generated in Valverde's Lab[19]. DDX3X myc/flag-tag was kindly provided by Dr. Hiroshi Yoshizawa from Niigata University. pEGFP-C1-DDX3X plasmid was kindly provided by Dr. Ming-Chih Lai from Chang Gung University. pEYFP-C1-DDX3X was generated by exchanging EGFP by EYFP sequences from Dr. Lai plasmid. Flag-tag Envelope protein form Zika virus was kindly provided by Dr. Tang from Howard University. BMV RNA2 was expressed from pB2NR3, a centromeric plasmid expressing 2a from a GAL promoter[57]. Plasmids to generate HCV (pFK-Luc-Jc1) and DENV (pFK-DVs-R2A) carrying the Firefly and Renilla luciferase reporter genes respectively have been previously described[58,59]. The plasmid to generate ZIKV carrying the reporter gene of NanoLuc luciferase (kindly provided by Dr. Merits, University of Tartu, Estonia) is based directly on the viral sequence isolated from a human patient from Brazil (isolate BeH819016 (KU365778.1). In transient knockdown experiments, control siRNA (5′-GCAGCACGACUUCUUCAAGdTdT-3′), TRPV4 siRNA (5′-CCAAGUUUGUUACCAAGAUtt-3′, Live Technologies) and DDX3X siRNA (5′-GAUGCUGGCUCGUGAUUUCUdTdT-3′, Eurogenetech) were used. Transient transfection was performed using Lipofectamine 2000.

**Cell cultures, transfection and cell lysis**. Mouse embryonic fibroblasts (MEF) were obtained from $Trpv4^{+/+}$ and $Trpv4^{-/-}$ mice[24]. Mice were provided by Prof. W. Liedtke (Duke University, Durham, NC, USA). HEK293 (catalog number 85120602) and HeLa (catalog number 93021013) cells were purchased from the European Collection of Authenticated Cell Cultures but not authenticated previous to their use in the present study. These cells were maintained in Dulbecco's modified Eagle's medium (DMEM, D6046 Sigma) supplemented with 10% fetal bovine serum, 100 units/ml penicillin and 100 units/ml streptomycin. HEK293 cells were used to preform co-immunoprecipitation assays because the high amount of protein obtained from transfected HEK293 cells. Human Bronchial Epithelial cell line (16HBE)[36] were obtained from Dr. D. C. Gruenert, San Francisco, CA, USA but not authenticated previous to their use in the present study. HBE cells were grown in modified Eagles medium with Earles salts (Gibco, Life Technologies), 10% fetal bovine serum (Gibco, Life Technologies), and 1% gentamicin (Sigma). The human hepatocarcinoma cell line Huh7/Scr (kindly provided by F. Chisari, The Scripps Research Institute, La Jolla, CA) was maintained in DMEM (Invitrogen, Carlsbad, CA) supplemented with 10% fetal bovine serum, 10% non-essential amino acids, 100 units/ml penicillin and 100 units/ml streptomycin. All cells were grown in an incubator with 5% $CO_2$ at 37 °C. Transfection was performed using polyethyleneimine (PEI, Polysciences, 23966). HEK293 cells overexpressing the transfected constructs were lysed 48 h after transfection and membrane proteins solubilized for 30 min at 4 °C in lysis buffer (50 mM Tris-HCl pH 7.4, 150 mM NaCl, 5 mM EDTA, 0.5% NP40, 1 mM DTT, 10 mM 13-GP, 0.1 mM $Na_3VO_4$, 1 μg/μl pepstatin, 2 μg/μl aprotinin, 0.1 mM PMSF, 1 mM benzamidine and EDTA-free protease inhibition cocktail, ROCHE 11873580001). Cell extracts were centrifuged at 14,000 × g at 4 °C for 10 min to remove aggregates.

**Pull down assay and protein identification by mass spectrometry**. TRPV4 and DDX3X-myc transfected HeLa cells were lysed and supernatant was incubated ON at 4 °C with anti Myc antibody followed by protein G beads (17-0618-01, GE) for 2 hours at 4 °C. The DDX3X containing beads were centrifuged, washed 3 times with PBS1X and subsequently digested with Trypsin (Promega, V5111) ON at 37 °C. Protein peptides were eluted from beads, desalted in a C18 stage tip (UltraMicronSpin Column, SUM SS18V) and loaded into a MAlDI-TOF-MS for protein identification. Full data obtained from mass-spectrometry analysis is provided as Supplementary Data 1.

**Co-immunoprecipitations**. Soluble fractions from cell lysis were used as input for co-immunoprecipitations. Cell extracts at 1 μg/μl (500 μg total protein) were

incubated ON at 4 °C with anti V5 antibody (R9612-25, Sigma). Immuno-complexes were then incubated with 50 μL of protein G beads (17-0618-01, GE) for 2 h at 4 °C. After incubation, complexes were washed with PBS1x buffer 3 times. Endogenous Co-IP was obtained from the lysis of ≈2.3 × 10⁸ Huh7 cells. Cells were lysed and proteins solubilized in lysis buffer. Cell extracts were centrifuged at 14000g at 4 °C for 10 min to remove aggregates. Solubilized proteins were incubated overnight with an anti-TRPV4 antibody generated against the final 20 aa of the human TRPV4[60]. Immuno-complexes were then incubated with 50 μL of protein G beads for 2 h at 4 °C. After incubation, complexes were washed with PBS 1× buffer 3 times. Immunoprecipitated complexes were denatured with SDS-PAGE sample buffer (90 °C for 5 min), separated by SDS-PAGE and analyzed by western blotting.

**Western blot**. Lysates and immunoprecipitates were loaded into SDS-page gels and run at 100mV for 90 min. For western blots of heterologously expressed proteins (Fig. 1a), the sample used for immunoprecipitation contains ten times more protein than the input sample. For western blots of native proteins (Fig. 1b), the sample used for immunoprecipitation contains hundred times more protein than the input. Gels were transferred to nitrocellulose membranes into an iBlot cast (IB1001, Invitrogen). Membranes were blocked in blocking buffer (5% non-fat-dry milk TTBS 1×) ON at 4 °C. Primary antibodies were incubated in blocking buffer for 1 h RT or overnight at 4 °C. Primary antibodies anti Lex A tag (sc-7544HRP, 200 μg/ml, SantaCruz), anti V5 tag (R9612-25, Sigma), anti Myc tag (M4439, 2 mg/ml stock, Sigma), anti-DDX3X (37160, 1 mg/ml stock, Abcam)[61], anti-TRPV4 (1.2 mg/ml stock, Live Technologies) and anti protein 2a (generated by the laboratory of P. Ahlquist, Wisconsin, USA)[62] were diluted 1:1000 in blocking buffer. Primary antibody anti PGK (22C5D8, 1 mg/ml stock, Molecular Probes) was diluted 1:4000 in blocking buffer. Anti-flag antibody (F1804, 1 mg/ml stock, Sigma) was used at a 1:1000 dilution. Secondary antibodies were incubated for 1 h at RT. Anti-mouse (NXA931, GE healthcare) and anti rabbit (NA934, GE healthcare) were used at a 1:2000 dilution in blocking buffer. Membranes were developed with West Pico or Femto (in case of protein 2a) chemiluminescent substrate (34080/34095, Thermo Fisher).

**Immunostaining and imaging**. Cells on coverslips were fixed in 4%PFA for 10 min at 24 hours post-transfection. Cells were then permeabilized in 0.1% Triton PBS 1× for 10 min and blocked in 2% BSA, 1% FBS for 1 hour. Antibodies were used in a 1:1000 dilution against TRPV4, DDX3X, Flag and Myc. Secondary antibodies A555 Mouse (A21424, Lifetechnologies), A647 Rabbit (A21245, Lifetechnologies), AlexaFluor 647 goat anti- rabbit IgG (A21244, 2 mg/ml stock, Invitrogen) and AlexaFluor 555 goat anti-mouse IgG (A212424, 2 mg/ml stock, Invitrogen) were used at a 1:2000 dilution. All antibodies were diluted in blocking buffer and incubated for 1 h. Nuclear staining was performed with Dapi (62248, Thermo-Fisher), diluted 1:1000 in PBS 1× for 10 min. Coverslips were mounted with Mowiol. Cells were examined with a Leica TCS-SP5 confocal microscope with a 63 × 1.40 immersion oil objective. We used the DAPI image for the identification of the nuclear region of interest to quantify DDX3X intensity.

**FRET measurements**. FRET measurements were performed using a Leica TCS SP5 confocal microscope and a ×63 oil immersion objective as previously described[19,33]. FRET efficiencies were expressed as the increase in FRET donor CFP after bleaching of the FRET acceptor YFP.

**Quantitative PCR**. Total RNA from cultured cells was extracted using the Nucleo-spin RNA isolation kit (740955, Macherey-Nagel) following manufacturer's protocol. RNA concentration was assessed using a NanoDrop ND 1000 Spectrophotometer (NanoDrop Technologies). cDNA synthesis was performed with 1 μg of total RNA using SuperScript III reverse transcriptase (18080093, Invitrogen). Quantitative PCR reactions (20 μl total volume), containing 1 μl cDNA template was performed using SYBR Green (4309155, Life technologies) with an ABI 7900HT real-time PCR machine (Applied Biosystems). Reactions were made using the following program: 50 °C for 2 min and 95 °C for 10 min, followed by 40 cycles of 95 °C for 15 s, 60 °C for 15 s and 72 °C for 15 s. cDNA reactions containing only water were used as negative controls in every experiment. The sequences of the gene-specific primers used are provided in Supplementary Table 2.

**HIV nuclear export and translation**. The plasmids pRev-mCherry, pMS2-YFP and p24XMBLGag-Pol-Vif-RRE were kindly provided by Dr. Nathan Sherer from University of Wisconsin-Madison[41]. pRev-mCherry encodes for Rev protein, that multimerizes on the RRE and binds the cellular factor chromosome region maintenance 1 (CRM1) to form a complex capable of exporting intron-containing HIV transcripts from the nucleus to the cytoplasm. pMS2-YFP encodes for MS2 protein, that is capable to bind to RNA MS2-binding loop (24xMBL), thus working as a reporter of RNAs containing the MBL. p24xMBL-Gag-Pol-Vif-RRE encodes for HIV-1 gRNAs containing RRE binding element for Rev transport, MS2-binding site (24xMBL) and expressing Gag-CFP protein. Huh7 cells were plated on coverslips and transiently transfected with the 3 reporter plasmids using polyethyleneimine (PEI, Polysciences, 23966). Huh7 cells were fixed with PFA 4% and stained with DAPI to mark nucleus. Cells were visualized under a Leica TCS-SP5 confocal microscope 36 hours post-transfection. The cells that were Rev-mCherry

positive were selected to measure MS2-YFP nuclear/total ratio and Gag-CFP intensity, which are directly proportional to HIV gRNAs nuclear export and HIV translation respectively.

**Ratiometric Ca²⁺ cell measurements**. Cytosolic $Ca^{2+}$ signals, relative to the fluorescence ratio (340/380) measured prior to cell stimulation, were obtained from transfected cells loaded with 4.5 mM fura-2 AM (Invirogen) as previously described[63]. Isotonic bath solutions used for $Ca^{2+}$ imaging experiments contained 140 mM NaCl, 2.5 mM KCl, 1.2 mM $CaCl_2$, 0.5 mM $MgCl_2$, 5 mM glucose, and 10 mM HEPES (pH 7.3). Hypotonic 30% bath solution was achieved by decreasing NaCl concentration by 30%.

**Zika virus calcium measurements**. Huh7 cells were seeded on a 24-well plate at a density of 70.000 cells/well. Cells were loaded with 4.5 μM Calbryte 520 AM (20650, AAT Bioquest, Inc.) for 30 min at 37 ºC. Cells were then washed in DMEM 1X without phenol red (11880-028, Gibco) twice. The measurements were performed in a FluoStar Optima Microplate Fluorimeter (BMGlabtech), with a 470 excitation filter and 520 emission filter. The fluorimeter was placed inside a cell culture laminar flow hood in a P2 biosafety room. Cells were incubated in 300 μl/ well of DMEM 1X without phenol red during the experiments. Zika virus was added at a MOI of 1 in DMEM 1× without phenol red. Wells containing only media were used to subtract the background signal.

**Generation of viral stocks**. Production of HCV virus from pFK-Luc-Jc1 and DENV from pFK-DVs-R2A plasmids have been previously described[64,65]. The plasmid carrying the ZIKV nanoLuc construct was linearized with the AgeI enzyme and purified with the QIAquick PCR purification kit (Qiagen, Düsseldorf, Germany). Purified DNA was subjected to a capping - in vitro transcription reaction with the mMessage mMachine SP6 kit (Ambion) according to the manufacturer's protocol. RNA from the in vitro transcription reaction was purified with the Nucleospin® RNA II kit (Macherey-Nagel, Düren, Germany). RNA integrity was verified by formaldehyde agarose gel electrophoresis and the concentration was determined by measurement of the optical density at 260 nm. For RNA electroporations, single-cell suspensions of Vero cells T100 cm² dishes were prepared by trypsinization. Cells were resuspended in 800 μl of phosphate-buffered saline (PBS) and mixed with 7 μg of in vitro transcribed RNA. Cells were immediately electroporated by using 3 pulses of 25 μF and 475 V in the Gene Pulser Xcell™ system (Bio-Rad, Munich, Germany) and a cuvette with a gap width of 0.4 cm (Bio-Rad). From each electroporation, three T100 cm² dishes were seeded.

For the generation of ZIKV stocks, supernatants of the electroporated cells were harvested at 3, 6 and 8 days post electroporation (filling with fresh media between harvestings), cleared by passing them through 45-μm-pore-size filters and stored at -80 ºC. For the determination of ZIKV viral titers a plaque assay was performed. Vero cells were seeded at a concentration of $8 \times 10^4$ cells/ml in a 12-well plate. Twenty-four hours later, serial dilutions of virus containing supernatant were added (2 wells per dilution) for 1h and later covered with a semi-solid solution of DMEM 2%+CMC. Five days later, the semi-solid solution was removed and every well was stained to visualize plaques.

**Yeast translation assay**. To study BMV RNA2 translation in the absence and presence of hTRPV4, WT (*Saccharomyces cerevisiae* Meyen ex E.C. Hansen (ATCC® 201388˜) and *ΔYvc1* (YSC1053 (ThermoFisherScientific) yeast strains were transformed with pB2NR3. The *ΔYvc1* strain was additionally transformed with pBT3-STE-TRPV4 or an empty plasmid. Cells were grown in 2% galactose and harvested in mid-log phase. Total protein was extracted and analyzed by western blot as previously described[66]. Total RNA was extracted and RNA2 and Act1 were analyzed by quantitative PCR (qPCR) using qScript XLT One-Step RT-qPCR ToughMix from Quanta Biosciences.

**Luciferase assay and cell cytotoxicity assay**. Standard infection assays were carried out as previously described[64]. Briefly, Huh7/Scr cells were seeded in 96-well plates with $1.2 \times 10^4$ cells/well. The following day, cells were preincubated for 1 h at 37 ºC with the different compounds and then inoculated with the different viruses and the compounds for 1 h in the case of ZIKV and 4 h in the case of HCV and DENV, at 37 ºC. Finally, the virus containing media was replaced with fresh media containing also the indicated compounds. Seventy-two hours post infection, Luciferase activity was assayed. Cells were washed with PBS, lysed in 150 μl of passive lysis buffer (for HCV) or Renilla lysis buffer (for DENV and ZIKV) and frozen. Upon thawing, lysates were resuspended by pipetting. A concentration of 50 μl of HCV-infected lysates were mixed with 25 μl of LARII for Firefly luciferase assays (Promega) and measured in a luminometer for 2 s. For DENV and ZIKV 4 μl of the lysates were mixed with 20 μl of Renila Luciferase Assay Buffer and 1/200 of substrate from the Renilla Luciferase assay system (Promega) and measured immediately in a luminometer for 2 s. Cytotoxicity (viability) was measured in all infection assays using the CytoTox-Glo cytotoxicity assay (Promega) as described by the manufacturer. Mean relative light units (RLU) were plotted as percentage relative to control infections (solvent without compounds) for both infectivity and cell viability. Half maximal effective concentration (EC50) and half maximal cytotoxic concentration 50 (CC50) were estimated by non-linear regression of log

inhibitor vs. normalized response and used to calculate the Selectivity Index (SI) value.

**Purification of protein E from Zika virus**. Approximately $5,5 \times 10^7$ HEK293 cells were transiently transfected with pcDNA3-Flag-Envelope Zika protein[67] (provided by Dr. Qiyi Tang, Howard University, Washington, USA). Cells were lysed 48 h post-transfection and protein E was solubilized in lysis buffer. Cell extracts were centrifuged at $14,000 \times g$ at 4 ºC for 10 min to remove aggregates. Solubilized proteins were incubated overnight with anti-Flag antibody (F1804, Sigma). Immuno-complexes were then incubated with 50 μL of protein G beads for 2 h at 4 ºC. After incubation complexes were washed with PBS 1× buffer 3 times. Immuno-complexes were eluted for 5 min with 100 μL 0.2 M glycine pH 2.5 and equilibrated with 20 μl of Tris-HCl 1 M, pH 8. Elution protocol was repeated to ensure complete elution. A fraction of purified protein E was denatured with SDS-PAGE sample buffer and analyzed by Coomassie staining and western blot.

**Statistical analysis**. The data was represented as mean ± SEM (unless otherwise indicated). Statistical analysis was assessed with Student's unpaired $t$ test, Mann–Whitney $U$ test, one-way analysis of variance (ANOVA) followed by Bonferroni post hoc test, or Kruskal–Wallis test followed by Dunn post hoc test, as suitable. In all cases a D'Agostino-Pearson omnibus normality test was performed prior to any hypothesis contrast test. Statistical analysis and graphics was performed using GraphPad. For the data that followed normal distributions, we applied either Student's $t$ test (when comparing two conditions) and one-way analysis of variance (ANOVA) followed by Tukey post hoc test (when comparing control with any other condition). For the data that did not fit a normal distribution, we used Mann–Whitney's unpaired $t$ test and non-parametric ANOVA (Kruskal–Wallis) followed by Dunn's post hoc test. The criterion for a significant difference was a final value of $P < 0.05$. * if $P < 0.05$, ** if $P < 0.01$ and *** if $P < 0.001$.

**Data availability**. The authors declare that the data supporting the findings of this study are available within the article and its Supplementary Information files, or are available from the authors upon request.

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

## Acknowledgements

We thank Dr. Andres Merits (University of Tartu, Estonia) for kindly providing the plasmid encoding the ZIKV with NanoLuc, Dr. Hiroshi Yoshizawa (Niigata University) for kindly providing the plasmid encoding DDX3X with Myc tag; and Dr. Dr. Ming-Chih Lai (Chang Gung University) for kindly providing the plasmid encoding DDX3X with GFP. This work was supported by the Spanish Ministry of Economy and Competitiveness through grants SAF2015-69762R, BFU2016-80039-R, BFU2017-87843-R, an institutional "Maria de Maeztu" Programme for Units of Excellence in R&D (MDM-2014-0370) and FEDER funds; Marie Curie International Outgoing Fellowship within the 7th European Community Framework Programme (PIOF-GA-2009-237120) and the Generalitat de Catalunya research program (AGAUR, 2014-SGR-1628 and FI-2013FIB00251).

## Author contributions

P.D-M. and M.A.V. conceived and designed the project. A.P-M designed the MYTH. P. D-M performed confocal imaging, MYTH, viral RNA export assays, calcium imaging

and Co-IP. F.R-M. helped with the calcium imaging experiments and generated the required molecular biology tools. J.D., J.J. and G.P-V. designed and performed viral infection assays and RNA2 translation assays. J.D. contributed to the interpretation of the data. M.A.V. wrote the manuscript. All authors edited the manuscript.

## Additional information

**Competing interests:** The authors declare no competing interests.

