## [Peer Review File · Nature Communications]

Reviewers' comments:

Reviewer #1 (Remarks to the Author):

In this manuscript, Donate-Macian et al. demonstrated that TRPV4, in a steady state, interacts with DDX3X, and after TRPV4 activation, DDX3X translocates into nucleus. Furthermore, DDX3X translocation is dependent on Ca²⁺ influx through TRPV4 and CaM/CaMKII pathway. Finally TRPV4 inhibition suppressed DDX3X-dependent viral RNA translation. Their findings that DDX3X translocates into nucleus in a TRPV4-activity-dependent manner, and depletion of TRPV4 decreased Dengue virus, Hepatitis C virus and Zika virus replication, are novel and potentially interesting. However, the functions of DDX3X are well established as they described in their manuscript, and physiological function of TRPV4 is still unclear because overall results are based on artificial experimental conditions. The manuscript could be improved as detailed below.

Specific comments

1. In figure 1A, they should put only TRPV4 sample to show the specific DDX3X-binding with TRPV4. Furthermore, to strengthen their data, they should perform reciprocal experiment (IP with DDX3X and IB with TRPV4).
2. In figure 1A, they describe that immunoprecipitated DDX3X were decreased by TRPV4 activation by GSK. However, I can also see the decreased TRPV4 protein expression in Input. Does GSK induce not only TRPV4 activation but also TRPV4 degradation? To strengthen their conclusion, they should quantify the immunoprecipitated DDX3X and would rather consider original TRPV4 protein expression.
3. In their manuscript, they investigated TRPV4-regulated mechanism and function in limited cell types. Therefore, physiological importance of TRPV4 is not so clear. To strengthen their conclusion, they should perform similar experiment using immune cells (e.g. primary macrophages or dendritic cells).
4. In figure 4A, they should perform statistical analysis to show the comparable expression of RNA2. Furthermore, they should quantify the 2A protein expression.
5. They should check whether virus infection really induces TRPV4 activation and Ca²⁺ influx through activated TRPV4. Furthermore, it is interesting to investigate whether TRPV4 mRNA and protein expression change in virally infected immune cells.
6. In figure 4, to clarify the importance of TRPV4 in immune responses, they should investigate the expression of inflammatory cytokines and type I interferons using immune cells.

Reviewer #2 (Remarks to the Author):

Comments to the authors:

The article entitled "The TRPV4 channel links calcium influx to DDX3X RNA helicase activity and viral infectivity" by Doñate-Macian and colleagues present preliminary evidence for a potential new regulatory role of TRPV4-mediated Ca²⁺ influx on the subcellular localization of the DEAD-box RNA helicase DDX3 and also on the potential role for TRPV4 activation in

RNA virus replication. Although DDX3X and TRPV4 were previously shown together in a β -arrestin 1 complex (Xiao 2007, PNAS), to my knowledge, this is the first report proposing a role of an ion channel and Ca^{+2} influx on DDX3X function. Despite authors certainly provide an interesting observation on the regulation of DDX3X localization/function upon TRPV4-mediated Ca^{+2} entry into the cell, my feeling is that the evidence presented concerning the consequences of this putative interaction is too preliminary and not sufficient to support author's claims. Thus, this manuscript should not be accepted for publication in Nature Communications in its current state.

Major comments:

One of my major concerns is regarding the experimental setting. Most data were obtained using overexpression of TRPV4 and DDX3X. Indeed, critical experiments showing TRPV4-DDX3X interaction as well as changes in the DDX3X subcellular localization upon TRPV4 activation were performed under overexpression conditions (authors do not justify the use of overexpression). These include experiments focused on the effects of the TRPV4 agonist GSK on DDX3X localization, which were performed in cells overexpressing both proteins. In this regard, there are logical questions that need to be solved:

- 1) Does the endogenous proteins indeed interact?
- 2) Can GSK treatment induce nuclear localization of the endogenous DDX3X?
- 3) Can GSK treatment induce nuclear localization of the endogenous DDX3X in the absence of overexpressed TRPV4?

Indeed, supplementary Figs. S3C and D show that neither GSK nor hypotonic media are able to induce (overexpressed) DDX3X relocation to the nucleus when TRPV4 is not overexpressed. This observation strongly suggests that the phenomenon described by the authors is dependent on high, non-physiological, levels of TRPV4.

Another important point is the quality of the images obtained by confocal microscopy, which are very important for the conclusions. For example, from Fig. 1B it seems that GSK treatment induce a decrease in the overall DDX3X signal. Moreover, from this figure and considering data presented in Figs 2A and 2C, one could expect some nuclear localization of DDX3X in the GSK condition. However, this is very difficult to observe from the figure since DAPI staining was not included. The same figure shows a "merge" between TRPV4 and DDX3 signals but it seems more likely that authors are showing "colocalization points" and not a merge. Information regarding image acquisition, processing and analysis was not provided (see below).

An additional problem of this manuscript is that it goes very fast and some important details on the experimental setting were not included. Some examples:

- Is the Co-PI image presented in Fig. 1A representative of several experiments?
- If yes, was any quantification of the TRPV4-bound DDX3X performed in order to support the idea that activation of the channel interferes with the TRPV4-DDX3X interaction? This is an important message of the manuscript but it seems that the conclusion was based only in

one single Co-IP experiment.

- How was Fig 1C obtained? Live-cell imaging using the EGFP-DDX3 vector described in the Methods section?
- How many cells were considered for the graph presented in Fig. 1D and how many independent experiments were performed?
- How was the nuclear/total ratio calculated? How many cells and how many independent experiments were performed? This point is also important considering the comments on the quality of the confocal microscopy images. Here, Western blots detecting DDX3X in nuclear and cytoplasmic fractions from control and GSK-treated cells will help to confirm author's claims.

Data presented in Fig. 4 are not conclusive since they are only indicative of a potential function of TRPV4 (as judged by the use of the specific inhibitor) on viral replication. Despite the rationale used to look for a role of TRPV4 on viral replication was based on its putative interaction with DDX3X (a known host factor required for RNA virus replication), none of the experiments presented in Fig. 4 support a role of TRPV4 on the activity of DDX3X on RNA metabolism or viral replication (as stated in the title of the manuscript). Hence, data presented in Fig. 4 are too preliminary.

Additional comments

- Fig. 2C should include a Western blot showing efficient knockdown of TRPV4.
- Fig. 4C shows mCherry-Rev localizing in the cytoplasm while this protein is mainly nuclear.
- "HIV-1 gRNAs" should read "HIV-1 gRNA" (HIV-1 genomic RNA).
- MOI of HCV, DENV and ZIKV used in experiments presented in Fig. 4F and 4G were not included in the figure legend nor in the Methods section.
- In Methods section, "94-well plate" should read "96-well plate"
- It is not clear how data were plotted. Were they normalized to the control, which was arbitrary set to 1 or 100%?

I hope these comments will help authors to improve the quality of their manuscript.

Reviewer #3 (Remarks to the Author):

The manuscript by Donate-Macian et al. used an unbiased yeast two-hybrid screening strategy to identify new intracellular signaling pathways regulated by TRPV4 cation channels. This approach revealed significant association of the channel with immune system response and viral infection. From these data, the authors selected DDX3X, an ATP-dependent RNA helicase, as the focus for the current study. The authors demonstrate that when overexpressed in HEK cells, TRPV4 and DDX3X can co-IP. Interestingly, co-IP was diminished by stimulating TRPV4 with the selective small molecule agonist GSK0106790A. Immunolabeling of HeLa cells for TRPV4 and DDX3X also suggests co-localization that was disrupted by GSK0106790A. The authors go on to show that activation of TRPV4 induces shuttling of DDX3X to the nucleus in a manner that is dependent on extracellular Ca²⁺.

However, increasing extracellular Ca²⁺ by other means, such as ionomycin or thapsigargin, did not induce translocation. A pharmacological approach indicates that calmodulin and CaMKIIb are involved with this response. Data demonstrating involvement of TRPV4 in viral replication are presented.

Overall, this is a very interesting story that demonstrates a new function for TRPV4 channels in viral infection. I have a few specific suggestions for improvement that I ask the authors to consider:

Specific Comments:

1. Figure 1B – these images are not terribly convincing. Basically, it looks like both TRPV4 and DDX3X are distributed throughout the cell, so it is not surprising that co-localization was detected. Further, the limit of resolution for confocal microscopy is at best 200 nm with is huge at the cellular level. Many techniques, such as FRET, PLA, and super-resolution microscopy are available that could significantly improve these data. Also, it is unclear if these studies and the corresponding the pull-down experiments were performed using an overexpression system. It is important to show this interaction in cells expressing TRPV4 and DDX3X at physiological levels.
2. It is difficult to understand how Ca²⁺ influx through TRPV4, but not global increases in intracellular Ca²⁺, stimulate translocation of DDX3X. The authors should at least discuss this finding in detail. It would be better if the authors explored this unexpected finding mechanistically.
3. The manuscript is rather terse, and in some sections, it is difficult to read because standard English is not used. I suggest that the authors expand the Introduction and include more discussion of their findings. A concluding section putting their findings into a larger perspective would greatly enhance the manuscript. Please edit the manuscript thoroughly.

RE: NCOMMS-17-28706

Doñate-Macian et al. "The TRPV4 channel links calcium influx to DDX3X RNA helicase activity and viral infectivity"

We are very grateful to all reviewers for their critical evaluation of our work. We have found all comments and suggestions enormously valuable. The answers to every point are included below

REVIEWER 1

Specific comments

1. In figure 1A, they should put only TRPV4 sample to show the specific DDX3X-binding with TRPV4. Furthermore, to strengthen their data, they should perform reciprocal experiment (IP with DDX3X and IB with TRPV4).

REPLY: Following the reviewer comments we have now included in Fig. S3 a Co-IP assay obtained from cells overexpressing only TRPV4(V5) and immunoprecipitating DDX3X with the anti-Myc antibody (reversed CoIP)

2. In figure 1A, they describe that immunoprecipitated DDX3X were decreased by TRPV4 activation by GSK. However, I can also see the decreased TRPV4 protein expression in Input. Does GSK induce not only TRPV4 activation but also TRPV4 degradation? To strengthen their conclusion, they should quantify the immunoprecipitated DDX3X and would rather consider original TRPV4 protein expression.

REPLY: We have now included CoIPs using cell lysates obtained from HuH7 cells that endogenously express TRPV4 and DDX3X (Fig. 1B). We also included the quantification of three CoIPs from cells overexpressing TRPV4 and DDX3X and one native CoIPs (Fig. 1C). Identical results were obtained in both conditions. The reviewer is right about the decreased presence of TRPV4 in the input of GSK treated cells. At times we have observed a decrease in the TRPV4 signal following channel activation. However, no differences in the immunoprecipitated TRPV4 could be detected between unstimulated and GSK stimulated cells, which is more relevant to the interpretation of the experiment. For the quantification of the CoIP we have corrected the amount Co-IP DDX3X by the total DDX3X in the input and the amount of precipitated TRPV4.

3. In their manuscript, they investigated TRPV4-regulated mechanism and function in limited cell types. Therefore, physiological importance of TRPV4 is not so clear. To strengthen their conclusion, they should perform similar experiment using immune cells (e.g. primary macrophages or dendritic cells)

REPLY: We have analyzed the TRPV4-DDX3X interaction in both heterologous expressing systems (yeast and HeLa cells) and cells endogenously expressing TRPV4 and DDX3X (HBE, Huh7 and MEF cells), as well as in cell systems in which the native proteins have been genetically silenced (yeast, MEF-TRPV4-KO or siRNA) or pharmacologically inhibited. In total, we have used five different cellular systems to analyze the physical interaction of these two proteins and its functional consequence in terms of RNA translation and viral infectivity. We agree that overexpression of modified proteins is an artificial condition imposed by the experimenter, but is typically complementary to other less "artificial" conditions and, at times, necessary to analyze biochemical and functional aspects that could not be approached by other means. With all our respect, we believe that the amount of data we have generated using all

these experimental models (6 in total, three of them endogenously expressing the proteins of interest) cannot be considered as the product of artificial conditions imposed on our experimental setup.

We have focused our study on epithelial cells capable of mounting innate responses (many of them typically use to evaluate viral infectivity) leaving immune cells out of the study.

Nevertheless, in an attempt to address the reviewer suggestion we analyze whether CY15 dendritic cell line and RAW macrophages present functional TRPV4 channels. None of these cell lines responded to GSK with increases of intracellular Ca^{2+} (see figure below), precluding further studies of the relevance of TRPV4 in the modulation of DDX3X activity.

4. In figure 4A, they should perform statistical analysis to show the comparable expression of RNA2. Furthermore, they should quantify the 2A protein expression.

REPLY. Both RNA2 levels (Fig. 5A, bottom) and protein 2A (Fig. 5A top) were quantified and expressed relative to the RNA2 levels to quantify protein 2A translation (Fig 5B). We apologies for not including the SEM and statistical analysis of RNA2 levels in our previous version. In this new version both SEM and stats are added to the figure.

5. They should check whether virus infection really induces TRPV4 activation and Ca^{2+} influx through activated TRPV4. Furthermore, it is interesting to investigate whether TRPV4 mRNA and protein expression change in virally infected immune cells.

REPLY.. We have now included data showing that both zika virus or purified ZIKV envelop protein induced an increase in intracellular Ca^{2+} concentration in Huh7 cells that was prevented by HC067047 (Fig. 6). As mentioned earlier, the focus of our study is not the immune system. So we have not performed any experiments with immune cells. We have no resident Ca-imaging setup in the biosafety cell culture room. So, in order to comply with the protocol of using scientific equipment (fluorimeter) that come in and out of the bio-safety cell culture room we used Zika virus for the live cell calcium measurements. We also tested whether virus infected Huh7 or MEF cells altered TRPV4 mRNA levels. Experiments addressing the expression of TRPV4 in response to viral infection that do not require the use of special equipment within the bio-safety room were carried out with both HCV and DNV. Neither viruses modified the expression of TRPV4 (Fig S10).

6. In figure 4, to clarify the importance of TRPV4 in immune responses, they should investigate the expression of inflammatory cytokines and type I interferons using immune cells.

REPLY: We tested whether TRPV4 activation or inhibition modified the expression of tumor necrosis factor α , interleukin-2, interleukin 1 β and interferon β in Huh7 cells. None of these changed under conditions that either activated or inhibited TRPV4 (Fig. S10).

REVIEWER 2

Comments to the authors:

“evidence presented concerning the consequences of this putative interaction is too preliminary and not sufficient to support author’s claims”

Major comments:

One of my major concerns is regarding the experimental setting. Most data were obtained using overexpression of TRPV4 and DDX3X. Indeed, critical experiments showing TRPV4-DDX3X interaction as well as changes in the DDX3X subcellular localization upon TRPV4 activation were performed under overexpression conditions (authors do not justify the use of overexpression). These include experiments focused on the effects of the TRPV4 agonist GSK on DDX3X localization, which were performed in cells overexpressing both proteins. In this regard, there are logical questions that need to be solved:

REPLY: We have analyzed the TRPV4-DDX3X interaction in both heterologous expressing systems (yeast and HeLa cells) and cells endogenously expressing TRPV4 and DDX3X (HBE, Huh7 and MEF cells; or yeast that express the orthologs Ded1 and Yvc), as well as in cell systems in which the native proteins have been genetically silenced (yeast, MEF-TRPV4-KO or siRNA). In total, we have used five different cellular systems to analysis the physical interaction of these two proteins and its functional consequence in terms of RNA translation and viral infectivity. We agree that overexpression of modified proteins is an artificial condition imposed by the experimenter, but is complementary to other experimental conditions and, at times, necessary to analyze biochemical and functional aspects that could not be approached by other means. We have now emphasized in the revised version all experiments carried out in cells endogenously expressing the proteins of interest. Therefore, we do not agree with the reviewer statement that our experiments are preliminary and mainly carried out on cells overexpressing TRPV4 and DDX3X.

1) Does the endogenous proteins indeed interact?

REPLY: We have now included CoIPs using cell lysates obtained from HuH7 cells that endogenously express TRPV4 and DDX3X (Fig. 1b). We also included the quantification of d 3 CoIPs from cells overexpressing TRPV4 and DDX3X and one native CoIPs (Fig. 1c). Identical results were obtained in both conditions.

2) Can GSK treatment induce nuclear localization of the endogenous DDX3X?

REPLY: Fig 2 shows the nuclear translocation of DDX3X following the activation of TRPV4 in three different cellular conditions:

- HeLa cells overexpressing TRPV4 and DDX3X (Fig 2a). HeLa cells do not express endogenous TRPV4 (Andrade J Cell Biol 2005)
- HBE cells endogenously express TRPV4 and DDX3X. To facilitate the nuclear tracking, DDX3X-myc was overexpressed (Fig. 2c).
- Huh7 cells that endogenously express both proteins. In this case we use an anti-DDX3X antibody to localize DDX3X (Fig 2c).

GSK only induced nuclear accumulation of DDX3X in the presence of TRPV4 (either endogenous or heterologously expressed). In summary, the functional interaction between DDX3X and TRPV4 has been evaluated in two cell lines endogenously expressing the proteins of interest and one cell line heterologously expressing these proteins.

3) Can GSK treatment induce nuclear localization of the endogenous DDX3X in the absence of overexpressed TRPV4? Indeed, supplementary Figs. S3C and D show that neither GSK nor hypotonic media are able to induce (overexpressed) DDX3X relocalization to the nucleus when TRPV4 is not overexpressed. This observation strongly suggests that the phenomenon described by the authors is dependent on high, non-physiological, levels of TRPV4

REPLY: HeLa Cells do not express endogenous TRPV4. Accordingly, HeLa cells transfected only with DDX3X did not respond with increases in intracellular Ca^{2+} concentration when challenged with hypotonic shocks or GSK (Fig S4A-B) and, no nuclear translocation of DDX3X occurred (Fig. S4C-D). We believe that the reviewer did not notice that HeLa cells do not express TRPV4. To avoid future misunderstanding by readers we have now clearly stated that HeLa cells do not express TRPV4.

Thus, we consider that the reviewer claim that our results depend on high, non-physiological levels of TRPV4 is incorrect. Both HBE and Huh7 cells (that endogenously expressed TRPV4 and DDX3X) were used throughout the study to characterize the DDX3X-TRPV4 physical and functional interaction.

4. Another important point is the quality of the images obtained by confocal microscopy, which are very important for the conclusions.

REPLY: We followed the reviewer suggestion and provide new images at higher magnification to better illustrate our claims. In new Fig. 1D we also modified the experimental conditions to highlight the TRPV4–DDX3X interaction at the plasma membrane. To remove excessive cytosolic signal cells were first permeabilized with digitonin, washed extensively to remove the cytosolic proteins, fixed, and then stained with the antibodies. This procedure revealed that a pool of DDX3X colocalized with TRPV4 (Fig. 1D). A clear overlapping of the signal plot profiles of TRPV4 and DDX3X was detected under control conditions at the plasma membrane that was reduced upon treatment with GSK1016790A (Fig. 1E).

For example, from Fig. 1B it seems that GSK treatment induce a decrease in the overall DDX3X signal. Moreover, from this figure and considering data presented in Figs 2A and 2C, one could expect some nuclear localization of DDX3X in the GSK condition.

REPLY: Although the original Fig. 1B is not included in the revised version we want to point out that the image showed that in the presence of GSK, DDX3X was clearly present in the nucleus (empty space seen at the centre of the cell in the TRPV4 staining image).

An additional problem of this manuscript is that it goes very fast and some important details on the experimental setting were not included. Some examples:

REPLY: we have carefully revised all sections of the manuscript to ensure all relevant information is included.

- Is the Co-PI image presented in Fig. 1A representative of several experiments?

REPLY: We have included the quantification of CoIPs from cells overexpressing TRPV4 and DDX3X and one native CoIP (Fig. 1C). Identical results were obtained in both conditions. Also

reverse ColP was included (Fig S3)

- How was Fig 1C obtained? Live-cell imaging using the EGFP-DDX3 vector described in the Methods section?

REPLY: Yes, we followed in vivo the colocalization of EGFP-DDX3X and ECFP-TRPV4.

- How many cells were considered for the graph presented in Fig. 1D and how many independent experiments were performed?

REPLY: 12 cells were monitored. Now included in the figure.

- How was the nuclear/total ratio calculated? How many cells and how many independent experiments were performed? This point is also important considering the comments on the quality of the confocal microscopy images. Here, Western blots detecting DDX3X in nuclear and cytoplasmic fractions from control and GSK-treated cells will help to confirm author's claims.

REPLY: Between three and six independent experiments were performed for each condition and the changes observed were consistent. The number of cells quantified per condition is shown for each column. Cells were examined with a Leica TCS-SP5 confocal microscope with a 63x 1.40 immersion oil objective. We used the DAPI image for the identification of the nuclear region of interest (ROI) to quantify nuclear DDX3X intensity. Total DDX3 was measured selecting a ROI of the whole cell (using always same gain and threshold) and measuring the DDX3 mean intensity signal within this ROI.

Data presented in Fig. 4 are not conclusive since they are only indicative of a potential function of TRPV4 (as judged by the use of the specific inhibitor) on viral replication. Despite the rationale used to look for a role of TRPV4 on viral replication was based on its putative interaction with DDX3X (a known host factor required for RNA virus replication), none of the experiments presented in Fig. 4 support a role of TRPV4 on the activity of DDX3X on RNA metabolism or viral replication (as stated in the title of the manuscript). Hence, data presented in Fig. 4 are too preliminary.

REPLY: New Fig 5 (old Fig 4) addresses the relevance of TRPV4-DDX3X interaction for well-known functions of DDX3X, particularly those related to viral RNA translation and multiplication. Ded1, the yeast orthologue of DDX3X, is required for translation of BMV RNA2 (Noueiry et al PNAS 2000). We show that compared to the WT yeast strain, depletion of the TRPV4 orthologue in yeast, *Yvc1* ($\Delta Yvc1$), significantly inhibited translation of RNA2, resulting in reduced levels of protein 2A (Fig. 5A, top) without modifying RNA2 levels (Fig. 5A, bottom). Transformation of the $\Delta Yvc1$ yeast strain with human TRPV4 recovered RNA2 translation to levels similar to those obtained in WT cells (Fig. 5A-B). These results clearly demonstrate that translation of a viral protein, that in yeast depends on the DDX3X ortholog Ded1, is controlled by the TRP channel, thereby providing a first clear link between TRPV4 and helicase activity. Second, the nuclear export of HIV RNA depends on DDX3X activity (Yedavalli et al 2004; Fröhlich et al 2016). We have now implemented the assay to monitor gRNA translocation and GAG translation (Fig. 5) with the silencing of DDX3X or TRPV4. Transfection of Huh7 cells with siDDX3X or siTRPV4 abrogated the effect of TRPV4 channel inhibition with HC on the nuclear export of gRNA and translation of Gag. These results confirmed that the effect of TRPV4 on viral RNA transport and translation relies on the presence of DDX3X. Finally, the infection by HCV, Zika and Dengue requires DDX3X and their infectivity is greatly reduced with the use of DDX3X inhibitors (Brai et al 2016), TRPV4 inhibition with HC067047 or

TRPV4 genetic knockout (Fig. 6). We have now included another TRPV4 inhibitor (RN1734) that also reduces HCV infectivity.

In summary, we have used five different techniques (biochemical and imaging) to show the interaction between DDX3X and TRPV4. We have shown that cellular localization of DDX3X is controlled by TRPV4-mediated Ca^{2+} influx triggered by physiological (hypotonic shocks), pharmacological (GSK) and pathological (virus infection) conditions. We have demonstrated that well-known functions of DDX3X related to viral RNA transport and replication are controlled by TRPV4 (and that in the absence of DDX3X such control by TRPV4 is lost). Finally we have demonstrated that infection of viruses that depend on DDX3X is reduced by either genetic silencing of TRPV4 or pharmacological inhibition using two different compounds. Because of all these solid pieces of evidence we cannot agree with the reviewer statement claiming that our data are too preliminary.

Additional comments

- Fig. 2C should include a Western blot showing efficient knockdown of TRPV4.

REPLY: To demonstrate the efficiency of the TRPV4 silencing, we have included qPCR of TRPV4 mRNA and Ca^{2+} -imaging to test the activity of TRPV4 in Huh7 cells transfected with siRNA-control and siTRPV4 at different time points after transfection (Fig S4H-J).

Fig. 4C shows mCherry-Rev localizing in the cytoplasm while this protein is mainly nuclear.

REPLY: Rev continuously shuttles between nucleus and cytosol. Similar to HeLa Cells where this system was first described, in our Huh7 model Rev shows a marked cytoplasmic localization. To test that inhibition of TRPV4 with HC067047 directly impacts on the gRNA nuclear export mediated by Rev we used a mutant REV (REV M10) that is retained in the nucleus. As expected, expression of REV M10 was exclusively detected in the nucleus whereas translation of GAG was totally abrogated (Fig. S8b). Under these conditions, treatment with HC067047 did not modify MS2 location (Fig. S8b-c) confirming that the effect of the channel inhibitor relied on the Rev-mediated nuclear export.

- "HIV-1 gRNAs" should read "HIV-1 gRNA" (HIV-1 genomic RNA).

Corrected

- MOI of HCV, DENV and ZIKV used in experiments presented in Fig. 4F and 4G were not included in the figure legend nor in the Methods section.

REPLY: This information is now included in the text.

- In Methods section, "94-well plate" should read "96-well plate"

Corrected

- It is not clear how data were plotted. Were they normalized to the control, which was arbitrary set to 1 or 100%?

YES.

REVIEWER 3

Specific Comments:

1. Figure 1B – these images are not terribly convincing. Basically, it looks like both TRPV4 and DDX3X are distributed throughout the cell, so it is not surprising that co-localization was detected. Further, the limit of resolution for confocal microscopy is at best 200 nm with is huge at the cellular level. Many techniques, such as FRET... Also, it is unclear if these studies and the corresponding the pull-down experiments were performed using an overexpression system. It is important to show this interaction in cells expressing TRPV4 and DDX3X at physiological levels.

REPLY: We have provided new images at higher magnification to better illustrate our claims. In new Fig. 1D we also modified the experimental conditions to highlight the TRPV4 –DDX3X interaction at the plasma membrane following a protocol we have previously used (Von Blume et al. Dev Cell 2011). To remove excessive cytosolic signal cells were first permeabilized with digitonin, washed extensively to remove the cytosolic proteins, fixed, and then stained with the antibodies. This procedure revealed that a pool of DDX3X colocalized with TRPV4 (Fig. 1D). A clear overlapping of the signal plot profiles of TRPV4 and DDX3X was detected under control conditions at the plasma membrane that was reduced upon treatment with GSK1016790A (Fig. 1E).

The proximity of TRPV4-CFP- and DDX3X-YFP-tagged proteins was also tested by the fluorescent resonance energy transfer (FRET) technique. Figure 1G shows a significant increase in maximal FRET efficiency in HeLa cells co-transfected with TRPV4-CFP and DDX3X-YFP compared to the control condition using soluble YFP or in the condition in which TRPV4 had been activated with GSK1016790A.

We also demonstrated the CoIP of TRPV4 and DDX3X, and the translocation of DDX3X in a more physiological cell system expressing endogenous DDX3x and TRPV4 (Fig. 1B-C and Fig 2C-D).

2. It is difficult to understand how Ca²⁺ influx through TRPV4, but not global increases in intracellular Ca²⁺, stimulate translocation of DDX3X. The authors should at least discuss this finding in detail. It would be better if the authors explored this unexpected finding mechanistically.

REPLY: Increases in Ca²⁺ concentration drive many cellular responses. The timing and location are key to coupling of the Ca²⁺ signal to the different cell responses. It is clear that nonuniformity and compartmentalization are essential features of the Ca²⁺ signals. This behavior calls for the presence of a scaffold that brings together the components required for a specific signaling module. The steep Ca²⁺ gradient around entry sites circumscribes the impact of Ca²⁺ signals to distances up to a few hundred Å. Thereby, downstream Ca²⁺-binding proteins should be in close proximity to the source of Ca²⁺. The TRPV4 cation channel may be well suited to act as a source of Ca²⁺ and the scaffold that brings together all elements of this specific signaling module. The interaction between TRPV4 and CaM has been previously reported and confirmed in our MYTH assay that also identified CaMKII and DDX3X as binding partners of TRPV4. Therefore, it is plausible that in the absence of TRPV4 no such scaffolding exists and the signaling module is not functional. Why Ca²⁺ increases generated by other ion transport pathways do not activate the CaM/CaMKII/DDX3X module? It may be that the global Ca²⁺ signals generated by ionomycin, thapsigargin or yoda1 are produced far away and do not reach the scaffolding microdomain. Alternatively, for the signal to progress it may be necessary a

coincident Ca^{2+} entry and TRPV4 conformational signaling triggered by the channel activation, as previously reported for voltage-gated Ca^{2+} channels (Li, Tadross & Tsien Science 2016). Future work will be required to decipher the molecular details that trigger this signaling module.

3. The manuscript is rather terse, and in some sections, it is difficult to read because standard English is not used. I suggest that the authors expand the Introduction and include more discussion of their findings. A concluding section putting their findings into a larger perspective would greatly enhance the manuscript. Please edit the manuscript thoroughly.

REPLY: We have thoroughly revised the manuscript including the reviewers suggestions. The manuscript has been revised by a native English speaker.

Reviewers' comments:

Reviewer #1 (Remarks to the Author):

The authors adequately addressed my concerns by adding new results. However, several points should be addressed.

1. It is a little bit strange that immunoprecipitated TRPV4 protein levels seem to be comparable between samples with GSK and without GSK while TRPV4 protein levels in Input are quite different (Figure 1B). They should optimize experimental conditions.
2. In figure 1C, they should clearly show which bar corresponds to which sample.

Reviewer #2 (Remarks to the Author):

Comments to the authors NCOMMS-17-28706

This reviewer acknowledges the efforts made by the authors to answer the comments raised during the first round of reviewing. The new version is clearer and provides additional data that strengthen the message and improves the quality of the manuscript.

Although I still have some minor concerns regarding the experimental setting and some conclusions, globally, I think that the findings presented regarding the role of the TRPV4 ion channel on the localization and function of DDX3X are well supported and will be of great interest for the scientific community as they open a new layer of regulation for this RNA helicase.

Some last minor comments:

There are no experiments directly demonstrating that the activation of the TRPV4 ion channel indeed affects the ATPase or RNA helicase activity of DDX3X (For example, authors have not used widely reported DDX3X mutants, such as the DQAD mutant, affecting these activities). Thus, I suggest that "RNA helicase" should be removed from the title. Authors should remember that DDX3X and other RNA helicases also have catalytic-independent functions.

Is intriguing that TRPV4 activation (which induces an increase in the nuclear localization of DDX3X), is required for efficient replication of cytoplasm-replicating viruses (ZIKV, HCV and DENV) that normally depend on cytoplasmic DDX3X. This is somehow contradictory and should be discussed. Indeed, the possibility that the function of TRPV4 on viral replication is not related to its interaction with DDX3X should not be discarded. Still, the role of TRPV4 on viral replication is itself very interesting.

Page 3 line 49: put capital letter in "transient"

Page 7 line 135: the term "real-time qPCR" is not correct when describing reverse

transcription (RT)-qPCR

Page 13 line 285: "whit" should read "with"

Page 14 line 302: remove comma from "(P12227,)"

Pages 17-18 lines 371, 377, 402: "coverlids" should read "coverslips"

Page 18 line 404: "Dapi" should read "DAPI"

Page 20 line 460: put capital letter in "western"

Page 21 line 468, 471 and 474: "ZIKAV" should read "ZIKV"

Reference 44 should be corrected.

The manuscript should benefit from the use of headings and subheadings in the Results section

Reviewer #3 (Remarks to the Author):

The authors have done a great job with the revision. I have no further comments.

REPLY to reviewers' comments.

Reviewer1.

1. It is a little bit strange that immunoprecipitated TRPV4 protein levels seem to be comparable between samples with GSK and without GSK while TRPV4 protein levels in Input are quite different (Figure 1B).

REPLY: The differences between input and immunoprecipitated signals in native Co-IP (Fig 1b) may be due to the difference in the amount of protein used for both techniques (much higher in the IP than in the WB of the input). Typically, in excess of protein compared to the antibody concentration is used in IP experiments. Nevertheless, the important point is that both conditions (+/- GSK) generated the same amount of immunoprecipitated TRPV4 and, therefore, is possible to compare the Co-IP of DDX3X. The following sentences have been added to the methods section to reflect these differences: " For Western blots of heterologously expressed proteins (Fig. 1a) the sample used for immunoprecipitation contains ten times more protein than the input sample. For Western blots of native proteins (Fig. 1b) the sample used for immunoprecipitation contains hundred times more protein than the input".

2. In figure 1C, they should clearly show which bar corresponds to which sample.

REPLY: Corrected

Reviewer 2.

1. I suggest that "RNA helicase" should be removed from the title.

REPLY: done

2. Is intriguing that TRPV4 activation (which induces an increase in the nuclear localization of DDX3X), is required for efficient replication of cytoplasm-replicating viruses (ZIKV, HCV and DENV) that normally depend on cytoplasmic DDX3X. This is somehow contradictory and should be discussed.

REPLY: The following sentences addressing this point have been added to the discussion: "In addition, TRPV4 appears to be required for the helicase-dependent translation of viral proteins (Fig. 7). However, it is not clear yet how TRPV4 activity links to the viral replication of cytoplasmic-replicating viruses such as ZIKV, HCV and DENV that mainly depend on cytoplasmic DDX3X, nor the relevance of the TRPV4-DDX3X interaction in such process".

3. All typos were corrected